# DEMYSTIFYING OVERCOMPLETE NONLINEAR AUTO-ENCODERS: FAST SGD CONVERGENCE TOWARDS SPARSE REPRESENTATION FROM RANDOM INITIALIZATION

## ABSTRACT

Auto-encoders are commonly used for unsupervised representation learning and for pre-training deeper neural networks. When its activation function is linear and the encoding dimension (width of hidden layer) is smaller than the input dimension, it is well known that auto-encoder is optimized to learn the principal components of the data distribution Oja (1982). However, when the activation is nonlinear and when the width is larger than the input dimension (overcomplete), auto-encoder behaves differently from PCA, and in fact is known to perform well empirically for sparse coding problems.

We provide a theoretical explanation for this empirically observed phenomenon, when rectified-linear unit (ReLu) is adopted as the activation function and the hidden-layer width is set to be large. In this case, we show that, with significant probability, initializing the weight matrix of an auto-encoder by sampling from a spherical Gaussian distribution followed by stochastic gradient descent (SGD) training converges towards the ground-truth representation for a class of sparse dictionary learning models. In addition, we can show that, conditioning on convergence, the expected convergence rate is $O(\frac{1}{t})$, where $t$ is the number of updates. Our analysis quantifies how increasing hidden layer width helps the training performance when random initialization is used, and how the norm of network weights influence the speed of SGD convergence.

## 1 INTRODUCTION

Let $x$ denote a vector in $\mathbb{R}^d$. An auto-encoder can be decomposed into two parts, encoder and decoder. The encoder can be viewed as a composition function $s_e \circ a_e : \mathbb{R}^d \to \mathbb{R}^n$; function $a_e : \mathbb{R}^d \to \mathbb{R}^n$ is defined as

$$a_e(x) := W_e x + b_e \text{ with } W_e \in \mathbb{R}^{n \times d}, b_e \in \mathbb{R}^n$$

$W_e$ and $b_e$ are the network weights and bias associated with the encoder. $s_e$ is a coordinate-wise activation function defined as

$$s_e(y)_j := s(y_j) \text{ where } s : \mathbb{R} \to \mathbb{R} \text{ is typically a nonlinear function}$$

The decoder takes the output of encoder and maps it back to $\mathbb{R}^d$. Let $x_e := s_e(a_e(x))$. The decoding function, which we denote as $\hat{x}$, is defined as

$$\hat{x}(x_e) := s_d(W_d x_e + b_d) \text{ with } W_d \in \mathbb{R}^{d \times n}, b_d \in \mathbb{R}^d, s_d : \mathbb{R}^d \to \mathbb{R}^d$$

where $(W_d, b_d)$ and $s_d$ are the network parameters and the activation function associated with the decoder respectively.

Suppose the activation functions are fixed before training. One can view $\hat{x}$ as a reconstruction of the original signal/data using the hidden representation parameterized by $(W_e, b_e)$ and $(W_d, b_d)$. The goal of training an auto-encoder is to learn the "right" network parameters, $(W_e, b_e, W_d, b_d)$, so that $\hat{x}$ has low reconstruction error.

**Weight tying**   A folklore knowledge when training auto-encoders is that, it usually works better if one sets $W_d = W_e^T$. This trick is called "weight tying", which is viewed as a trick of regularization, since it reduces the total number of free parameters. With tied weights, the classical auto-encoder is simplified as

$$\hat{x}(s_e(a_e(x))) = s_d(W^T s_e(Wx + b_e) + b_d)$$

In the rest of the manuscript, we focus on weight-tied auto-encoder with the following specific architecture:

$$\hat{x}_{W,b}(x) = W^T s_{ReLu}(a(x)) = W^T s_{ReLu}(Wx + b) \ \text{ with } \ s_{ReLu}(y)_i := \max\{0, y_i\} \qquad (1)$$

Here we abuse notation to use $\hat{x}_{W,b}$ to denote the encoder-decoder function parametrized by weights $W$ and bias $b$. In the deep learning community, $s_{ReLu}$ is commonly referred to as the rectified-linear (ReLu) activation.

**Reconstruction error**   A classic measure of reconstruction error used by auto-encoders is the expected squared loss. Assuming that the data fed to the auto-encoder is i.i.d distributed according to an unknown distribution, i.e., $x \sim p(x)$, the population expected squared loss is defined as

$$L(W, b) := \frac{1}{2} E_{x \sim p(x)} \|x - \hat{x}_{W,b}(x)\|^2 \qquad (2)$$

Learning a "good representation" thus translates to adjusting the parameters $(W, b)$ to minimize the squared loss function. The implicit hope is that the squared loss will provide information about what is a good representation. In other words, we have a certain level of belief that the squared loss characterizes what kind of network parameters are close to the parameters of the latent distribution $p(x)$. This unwarranted belief leads to two natural questions that motivated our theoretical investigation:

- Does the global minimum (or any of global minima, if more than one) of $L(W, b)$ correspond to the latent model parameters of distribution $p(x)$?

- From an optimization perspective, since $L(W, b)$ is non-convex in $W$ and is shown to have exponentially many local minima Safran & Shamir (2016), one would expect a local algorithm like stochastic gradient descent, which is the go-to algorithm in practice for optimizing $L(W, b)$, to be stuck in local minima and only find sub-optimal solutions. Then how should we explain the practical observation that auto-encoders trained with SGD often yield good representation?

**Stochastic-gradient based training**   Stochastic gradient descent (SGD) is a scalable variant of gradient descent commonly used in deep learning. At every time step $t$, the algorithm evaluates a stochastic gradient $g(\cdot)$ of the population loss function with respect to the network parameters using back propagation by sampling one or a mini-batch of data points. The weight and bias update has the following generic form

$$W^{t+1} \leftarrow W^t - \eta_w^t g^{t+1}(W^t), \ \text{ with } \ Eg(W^t) = \frac{\partial L(W, b)}{\partial W}(W^t)$$

$$b^{t+1} \leftarrow b^t - \eta_b^t g^{t+1}(b^t) \ \text{ with } \ Eg(b^t) = \frac{\partial L(W, b)}{\partial b}(b^t)$$

where $\eta_w^t$ and $\eta_b^t$ are the learning rates for updating $W$ and $b$ respectively, typically set to be a small number or a decaying function of time $t$. The unbiased gradient estimate $g(W^t)$ and $g(b^t)$ can be obtained by differentiating the empirical loss function defined on a single or a mini-batch of size $m$, denoted by $\ell_m$:

$$\ell_m(W, b) := \frac{1}{m} \sum_{i=1}^{m} \ell(W, b; x_i), \ \text{ where } \ \ell(W, b; x) := \frac{1}{2} \|x - \hat{x}_{W,b}(x)\|^2 \qquad (3)$$

Then the stochastic or mini-batch gradient descent update can be written as

$$W^{t+1} \leftarrow W^t - \eta_w^t \frac{\partial \ell_m(W, b)}{\partial W}(W^t) \ \text{ and } \ b^{t+1} \leftarrow b^t - \eta_b^t \frac{\partial \ell_m(W, b)}{\partial b}(b^t) \qquad (4)$$

Table 1: Organization of notations: the "parameters" are those whose value determine the performance guarantee of auto-encoders; the "auxiliary" variables are only used to facilitate our analysis.

| Model | | Algorithm | | Analysis |
| --- | --- | --- | --- | --- |
| parameters | auxiliary | parameters | auxiliary | auxiliary |
| $k$(dictionary size) | $x$ | $c$ (norm control) | $W^t$ | $\delta$ |
| $d$ (dimension) | $W^*$ | $c'$ (learning rate | $b^t$ | $\tau_{s,1}, \tau_{s,2}$ |
| $\lambda$ (incoherence) | $C_j$ | $t_o$ parameters) | $a^t(\cdot)$ | $g(\cdot)$ |
| $\epsilon, \sigma$ (noise) | | $n$ (width of hidden layer) | | |

**Max-norm regularization**    A common trick called "max-norm regularization" Srivastava et al. (2014) or "weight clipping" is used in training deep neural networks. [1] In particular, after each step of stochastic gradient descent, the updated weights is forced to satisfy

$$\max_i \|W_{i,\star}\|_2 \le c$$

for some constant $c$. This means the row norm of the weights can never exceed the prefixed constant $c$. In practice, whenever $\|W_{i,\star}\|_2 > c$, the max-norm constraint is enforced by projecting the weights back to a ball of radius $c$.

## 2    PRELIMINARIES

In this section, we start by defining notations. Then we introduce a norm-controlled variant of SGD algorithm that operates on the auto-encoder architecture formalized in (1). Finally, we introduce assumptions on the data generating model.

**General principle of notations**    We use the same notation for network parameters $W, b$, and for activation $a(\cdot)$, as in Section 1. We use $s(\cdot)$ as a shorthand for the ReLu activation function $s_{ReLu}(\cdot)$. We use capital letters, such as $W$ or $F$, either to denote a matrix or an event; we use lower case letters, such as $x$, for vectors. $W^T$ denotes the transpose of $W$. We use $W_{s,\star}$ to denote the $s$-th row of $W$. When a matrix $W$ is modified through time, we let $W^t$ denote the state of the matrix at time $t$, and $W^t_{s,\star}$ for the state of the corresponding row. We use $\|\cdot\|$ for $l_2$-norm of vectors and $|\cdot|$ for absolute value of real numbers. Matrix-vector multiplication between $W$ and $x$ (assuming their dimensions match) is denoted by $Wx$. Inner product of vectors $x$ and $y$ is denoted by $\langle x, y \rangle$.

**Organization of notations**    Throughout the manuscript, we introduce notations that can be divided into "model", "algorithm", and "analysis" categories according to their utility. They are organized in Table 1 to help readers interpreting our results. For example, If a reader is interested in knowing how to apply our result to parameter tuning in training auto-encoders, then she might ignore the auxiliary notations and only refer to algorithmic parameters and model parameters in Table 1, and examine how does the setting of the former is influenced by the latter in Theorem 1.

### 2.1    NORM-CONTROLLED SGD TRAINING

We assume that the algorithm has access to i.i.d. samples from an unknown distribution $p(x)$. This means the algorithm can access stochastic gradients of the population squared-loss objective in (2) via random samples from $p(x)$. The norm-controlled SGD variant we analyze is presented in Algorithm 1 (it can be easily extended to the mini-batch SGD version, where for each update we sample more than one data points). It is almost the same as what is commonly used in practice: it random initializes the weight matrix by sampling unit spherical Gaussian, and at every step the algorithm moves towards the direction of the negative stochastic gradient with a linearly decaying learning rate.

However, there are two differences between Algorithm 1 and original SGD: first, we impose that the norm of the rows of $W^t$ be controlled; this is akin to the practical trick of "max-norm regularization"

---

[1] The name max-norm regularization was originally proposed as a technique for low-rank matrix factorization Srebro & Shraibman (2005), where the definition is in fact not exactly the same as what is practiced in the deep learning community. We use the latter convention in our analysis.

as explained in Section 1; second the update of bias is chosen differently than what is usually done in practice, which deserves additional explanation.

**Comment on the setting of bias in Algorithm 1**    The stochastic gradient of bias $b$ with respect to squared loss in (2) can be evaluated by sampling a single data point and differentiate against the empirical loss in (3), can be derived as

$$\frac{\partial \ell(W, b; x)}{\partial b_j} = -\frac{\partial s(a_j)}{\partial a_j} \langle r, W_{j\star} \rangle \quad \text{(derivation can be found in (6) of the Appendix)}$$

Since the gradient is noisy, the generic form of SGD suggests modifying $b_j^t$ using the update

$$b_j^{t+1} \leftarrow b_j^t + \eta_b^t \frac{\partial s(a_j)}{\partial a_j} \langle r, W_{j\star} \rangle$$

for a small learning rate $\eta_b^t$ to mitigate noise. This amounts to stepping towards the negative gradient direction and move a little. On the other hand, we can directly find the next update $b_j^{t+1}$ as the point that sets the gradient to zero, that is, we find $b_j^*$ such that

$$\frac{\partial \ell(W^t, b^t; x')}{\partial b}(b_j^*) = 0$$

The closed form solution to this is to choose

$$b_j^* = \langle x' 1_{\{a^t(x') > 0\}}, W_{j\star}^t \rangle (\frac{1}{\|W_{j\star}^t\|^2} - 1)$$

This strategy, which is essentially Newton's algorithm, should perform better than gradient descent if we have an accurate estimate of the true gradient, so it would likely benefit from evaluating the gradient using a mini-batch of data. If, on the other hand, the gradient is very noisy, then this method will likely not work as well as the original SGD update. Analyzing the evolvement of both $W^t$ and $b^t$, which has dependent stochastic dynamic if we follow the original SGD update, would be a daunting task. Thus, to simplify our analysis, we assume in our analysis that we have access to

$$E_x \langle x' 1_{\{a^t(x') > 0\}}, W_{j\star}^{t+1} \rangle (\frac{1}{\|W_{j\star}^{t+1}\|^2} - 1)$$

The substitute of $W_{j\star}^{t+1}$ for $W_{j\star}^t$ is to further simplify our analysis. In practice, this update can be implemented by first updating $W_{j\star}^t$ to $W_{j\star}^{t+1}$, and then updating $b_j^t$ using $W_{j\star}^{t+1}$.

## 2.2   A SIMPLE SPARSE DICTIONARY LEARNING MODEL

We assume that the data $x$ we sample follows the dictionary learning model

$$x = (W^*)^T s + \epsilon$$

where $W^* \in \mathbb{R}^{k \times d}$. Here $k$ is the size of the dictionary, which we assume to be at least two (otherwise, the model becomes degenerate), and the true value of $k$ is **unknown to the algorithm**.

The rows of $W^*$ are the dictionary items; $W_{j\star}^*$ satisfies

$$\|W_{j\star}^*\| = 1, \forall j \in [k]$$

Let the incoherence between dictionary items be defined as $\lambda := \max_{j, i \neq j, i, j \in [k]} |\langle W_{j\star}^*, W_{i\star}^* \rangle|$, we assume that $\lambda \leq \frac{1}{8k}$. In our simplified model, the coefficient vector $s \in \{0, 1\}^k$ is assumed to be 1-sparse, with

$$Pr(s_j = 1) = \frac{1}{k}$$

$$E\epsilon = 0 \text{ and } E[\epsilon\epsilon^T] = \sigma^2 I \text{ with } \sigma^2 \leq \frac{\lambda}{2\sqrt{2}d}.$$

Finally, we assume that the noise has bounded norm [2]: $\max \|\epsilon\| \leq \frac{\sqrt{1-\lambda^2}}{6k}$.

---

[2]This assumption can be easily relaxed to a probabilistic bound, e.g., by assuming the noise has sub-gaussian tails. We stick with this stronger assumption for simplicity

---

**Algorithm 1** Norm-controlled SGD training

---

**Input:** width parameter $n$; norm parameter $c$; learning rate parameters $c', t_o, \delta$; total number of iterations, $t_{\max}$.

**Initialization of $W^o$:** For all $s \in [n]$,

$\quad W^o_{s\star} \leftarrow c\frac{z}{\|z\|}$, where $z \in \mathbb{R}^d$, $z_i \sim N(0,1)$

**Initialization of $b^o$:** Sample $x \sim p(x)$; for all $s \in [n]$,

$\quad$ Find $b^*_s$ such that $\frac{\partial \ell(x; W^o, \vec{0})}{\partial b}(b^*_s) = 0$

$\quad b^o_s \leftarrow b^*_s$

$\quad$ (version used in analysis: $b^o_s \leftarrow E_x \langle x, W^o_{s\star} \rangle (\frac{1}{c^2} - 1)$)

**while** $t \leq t_{\max}$ **do**

$\quad\quad W^{t+1} \leftarrow W^t - \eta^t \frac{\partial \ell(x, W^t, b^t)}{\partial W}$, where $x \sim p(x)$

$\quad\quad W^{t+1}_{j\star} \leftarrow \frac{W^{t+1}_{j\star}}{\|W^{t+1}_{j\star}\|}$

$\quad\quad$ Draw a fresh sample $x' \sim p$; for all $s \in [n]$,

$\quad\quad\quad$ Find $b^*_s$ such that $\frac{\partial \ell(x'; W^t, b^t)}{\partial b}(b^*_s) = 0$

$\quad\quad\quad b^{t+1}_s \leftarrow b^*_s$, or equivalently, $b^{t+1}_s \leftarrow \langle x' 1_{\{a^t(x') > 0\}}, W^{t+1}_{s\star} \rangle (\frac{1}{c^2} - 1)$

$\quad\quad\quad$ ( version used in analysis $b^{t+1}_s \leftarrow E_x \langle x' 1_{\{a^t(x') > 0\}}, W^{t+1}_{s\star} \rangle (\frac{1}{c^2} - 1)$ )

**end while**

**Output:** $W^{t_{\max}}, b^{t_{\max}}$

---

While auto-encoders are often related to PCA, the latter cannot reveal any information about the true dictionary under this model even in the complete case, where $d = k$, due to the **isotropic** property of the underlying distribution.

The data generating model can be equivalently viewed as a mixture model: for example, when $s_j = 1$, it means $x$ is of the form $W^*_{j\star} + \epsilon$. When $\epsilon$ is Gaussian, the model coincides with mixture of Gaussians model, with the dictionary items being the latent locations of individual Gaussians. Thus, we adopt the concept from mixture models, and use $x \sim C_j$ to indicate that $x$ is generated from the $j$-th component of the distribution.

## 3 MAIN RESULTS

To formally study the convergence property of Algorithm 1, we need a measure to gauge the distance between the learned representation at time $t$, $W^t$, and the ground-truth representation, $W^*$, which may have different number of rows. There are potentially different ways to go about this. The distance measure we use is

$$\Theta(W^t, W^*) := \frac{1}{k} \sum_{j \in [k]} \min_{s \in [n]} \Delta(W^t_{s\star}, W^*_{j\star}) \text{ with } \Delta(W^t_{s\star}, W^*_{j\star}) := 1 - (\langle \frac{W^t_{s\star}}{\|W^t_{s\star}\|}, W^*_{j\star} \rangle)^2$$

Note that $\Delta(W^t_{s\star}, W^*_{j\star})$ is the squared sine of the angle between the two vectors, which decreases monotonically as their angle decreases, and equals zero if and only if the vectors align. Thus, $\min_{s \in [n]} \Delta(W^t_{s\star}, W^*_{j\star})$ can be viewed as the angular distance from the best approximation in the learned hidden representations of the network, to the ground-truth dictionary item $W^*_{j\star}$. And $\Theta(\cdot, \cdot)$ measures this distance averaged over all dictionary items.

Our main result provides recovery and speed guarantee of Algorithm 1 under our data model.

**Theorem 1.** *Suppose we have access to i.i.d. samples $x \sim p(x)$, where the distribution $p(x)$ satisfies our model assumption in Section 2.2. Fix any $\delta \in (0, \frac{n}{e})$. If we train auto-encoder with norm-controlled SGD as described in Algorithm 1, with the following parameter setting*

- *The row norm of weights set to be $\|W^t_{s\star}\| = c$ ($\forall s \in [n], \forall t$) such that $\frac{3}{2} \leq c \leq \sqrt{6k}$*

- *If the bias update at $t$ is chosen such that*

$$b^{t+1}_s = E_x \langle x' 1_{\{a^t(x') > 0\}}, W^{t+1}_{s\star} \rangle (\frac{1}{c^2} - 1)$$

- *The learning rate of SGD is set to be $\eta^t := \frac{c'}{t+t_o}$, with $c' > 2kc$ and $t_o \geq \frac{192(c')^2 B^2}{\lambda^2}(\ln \frac{n}{\delta})^2$*

*Then Algorithm 1 has the following guarantees*

- *When random initialization with i.i.d. samples from $N(0,1)$ is used, the algorithm will be initialized successfully (see definition of successful initialization in Definition 1) with probability at least $1 - k \exp\{-n(\frac{\lambda}{\sqrt{2}})^{d-3}\}$.*

- *When random initialization with i.i.d. samples $x \sim p(x)$ is used, the algorithm will be initialized successfully with probability at least*

$$(1 - k \exp\{-\frac{n\lambda^2}{8kB}\})(1 - 3\exp\{-\frac{n^3}{100k^2}\})$$

- *Conditioning on successful initialization, let $\Omega$ denote the sample space of all realizations of the algorithm's stochastic output, $(W^1, W^2, \dots, )$. Then at any time $t$, there exists a large subset of the sample space, $F^t \subset \Omega$, with $Pr(F^t) \geq 1 - \delta$, such that*

$$E[\Theta(W^t, W^*)|F^t] \leq (\frac{t_o + 1}{t_o + t + 1})^4 \frac{\lambda^2}{2} + \frac{(c')^2 B}{3}(1 + \frac{1}{t_o + 1})^{\frac{2c'}{kc}+1} \frac{1}{t_o + t + 1}$$

**Interpretation** The first statement of the theorem suggests that the probability of successful initialization increases as the width of hidden layer increases. In particular, when Gaussian initialization is used, in order to ensure a significantly large probability of successful initialization, the analysis suggests that the number of neurons required must scale as $\Omega(\lambda^{-d}) = \Omega(k^d)$, which is exponential in the ambient dimension. When the neurons are initialized with samples from the unknown distribution, the analysis suggests that the number of neurons required scale as $\Omega(\frac{k}{\lambda^2}) = \Omega(k^3)$, which is polynomial in the number of dictionary size. Hence, our analysis suggests that, at least under our specific model, initializing with data is perhaps a better option than Gaussian initialization. The second statement suggests that conditioning on a successful initialization, the algorithm will have expected convergence towards $W^*$, measured by $\Theta(\cdot, \cdot)$, of order $O(\frac{1}{t})$. If we examine of form of bound on the convergence rate, we see that the rate will be dominated by the second term, whose constant is heavily influenced by the choice of learning rate parameter $c'$.

**Explaining distributed sparse representation via gradient-based training** The main advantage of gradient-based training of auto-encoders, as revealed by our analysis, is that it simultaneously updates all its neurons in parallel, in an independent fashion. During training, a subset of neurons will specialize at learning a single dictionary item: some of them will be successful while others may fail to converge to a ground-truth representation. However, since the update of each neuron is independent (in an algorithmic sense), when larger number of neurons are used (widening the hidden layer), it becomes more likely that each ground-truth dictionary will be learned by some neuron, even from random initialization.

## 4 RELATED WORKS

Despite the simplicity of auto-encoders in comparison to other deep architectures, we still have a very limited theoretical understanding of them. For linear auto-encoders whose width $n$ is less than than its input dimension $d$, the seminal work of Oja (1982) revealed their connection to online stochastic PCA. For non-linear auto-encoders, recent work Arpit et al. (2016) analyzed sufficient conditions on the activation functions and the regularization term (which is added to the loss function) under which the auto-encoder learns a sparse representation. Another work Rangamani et al. (2017) showed that under a class of sparse dictionary learning model (which is more general than ours) the ground-truth dictionary is a critical point (that is, either a saddle point or a local miminum) of the squared loss function, when ReLu activation is used. We are not aware of previous work providing global convergence guarantee of SGD for non-linear auto-encoders, but our analysis techniques are closely related to recent works Balsubramani et al. (2013); Ge et al. (2015); Tang & Monteleoni (2017) that are at the intersection of stochastic (non-convex) optimization and unsupervised learning.

**PCA, $k$-means, and sparse coding** The work of Balsubramani et al. (2013) provided the first convergence rate analysis of Oja's and Krasulina's update rule for online learning the principal component (stochastic 1-PCA) of a data distribution. The neural network corresponding to 1-PCA has a single node in the hidden layer without activation function. We argue that a ReLu activated width $n$ auto-encoder can be viewed as a generalized, multi-modal version of 1-PCA. This is supported by our analysis: the expected improvement of each neuron, $W_s^t$, bears a striking similarity to that obtained in Balsubramani et al. (2013). The training of auto-encoders also has a similar flavor to online/stochastic $k$-means algorithm Tang & Monteleoni (2017): we may view each neuron as trying to learn a hidden dictionary item, or cluster center in $k$-means terminology. However, there is a key difference between $k$-means and auto-encoders: the performance of $k$-means is highly sensitive to the number of clusters. If we specify the number of clusters, which corresponds to the network width $n$ in our notation, to be larger than the true $k$, then running $n$-means will over-partition data from each component, and each learned center will not converge to the true component center (because they converge to the mean of the sub-component). For auto-encoders, however, even when $n$ is much larger than $k$, the individual neurons can still converge to the true cluster center (dictionary item) thanks to the independent update of neurons. SGD training of auto-encoders is perhaps closest to a family of sparse coding algorithms Schnass (2015); Arora et al. (2015). For the latter, however, a critical hyper-parameter to tune is the threshold at which the algorithm decides to cut off insignificant signals. Existing guarantees for sparse coding algorithms therefore depend on knowing this threshold. For ReLu activated auto-encoders, the threshold is adaptively set for each neuron $s$ at every iteration as $-b_s^t$ via gradient descent. Thus, they can be viewed as a sparse coding algorithm that self-tunes its threshold parameter.

## 5 ANALYSIS

In our analysis, we define an auxiliary variable

$$\phi(W_{s\star}^t, W_j^*) := 1 - \Delta(W_{s\star}^t, W_j^*)$$

Note that $\phi(\cdot, \cdot)$ is the squared cosine of the angle between $W_{s\star}^t$ and $W_j^*$, which increases as their angle decreases. Thus, $\phi$ can be thought as as measuring the angular "closeness" between two vectors; it is always bounded between zero and one and equals one if and only if the two vectors align.

Our analysis can be divided into three steps. We first define what kind of initialization enables SGD to converge quickly to the correct solution, and show that when the number of nodes in the hidden layer is large, random initialization will satisfy this sufficient condition. Then we derive expected the per-iteration improvement of SGD, conditioning on the algorithm's iterates staying in a local neighborhood (Definition 4). Finally, we use martingale analysis to show that the local neighborhood condition will be satisfied with high probability. Piecing these elements together will lead us to the proof of Theorem 1, which is in the Appendix.

### 5.1 PART I: PERFORMANCE GUARANTEE OF INITIALIZATION

**Covering guarantee from random initialization** Intuitively, for each ground-truth dictionary item, we only require that at least one neuron is initialized to be not too far from it.

**Definition 1.** *If the rows of $W^o$ have fixed norm $c > 0$. Then we define the event of **successful initialization** as*

$$F_*^o := \left\{ \min_{j \in [k]} \max_{i \in [n]} \langle W_{i\star}^o, W_{j\star}^* \rangle \geq c\sqrt{1 - \frac{\lambda^2}{2}} \right\}$$

**Lemma 1** (Random initialization with Gaussian variables). *Suppose $W^o \in \mathbb{R}^{n \times d}$ is constructed by drawing $z_{i,j} \sim N(0, 1)$ for all $i \in [n], j \in [d]$, and setting $W_{i,\star}^o = c\frac{z_{i\star}}{\|z_{i\star}\|}$. Then*

$$Pr\{F_*^o\} \geq 1 - k \exp\{-n(\frac{\lambda}{\sqrt{2}})^{d-3}\}$$

**Lemma 2** (Random initialization with data points). *Suppose $W^o \in \mathbb{R}^{n \times d}$ is constructed by drawing $X_1, \ldots, X_n$ from the data distribution $p(x)$, and setting $W_{i,j}^o = c\frac{X_i}{\|X_i\|}$, for all $i \in [n]$. If $\sigma^2 \leq \frac{\lambda}{2\sqrt{2}d}$,*

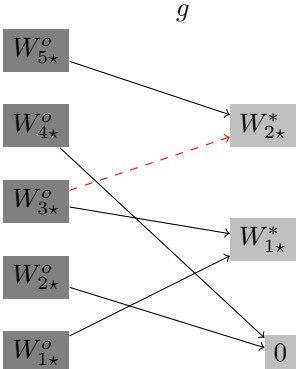

Figure 1: The auto-encoder in this example has 5 neurons in the hidden layer and the dictionary has two items; in this case, $g(1) = g(3) = 1$, $g(5) = 2$, and the other two neurons do not learn any ground-truth (neurons mapped to 0 are considered useless). Under unique firing condition, which holds when the dictionary is sufficiently incoherent, the red dashed connection will not take place (each neuron is learning at most one dictionary item).

*then*

$$Pr\{F_*^o\} \geq (1 - k\exp\{-\frac{n\lambda^2}{8kB}\})(1 - 3\exp\{-\frac{n^3}{100k^2}\})$$

**Definition 2.** *Conditioning on $F^o$, we can map the rows of $W^o$ to an dictionary item $W_{j\star}^*$, $j \in [k]$, according to the following **firing map** [3]*

$$g : [n] \rightarrow \{0, 1, \ldots, k\} \ s.t. \ \begin{cases} g(s) = j \ if \ \langle W_{s\star}^o, W_{j\star}^* \rangle \geq c\sqrt{1-\lambda^2} \\ g(s) = 0 \ otherwise \end{cases}$$

Figure 1 provides an illustration of firing map $g(\cdot)$. Note that some rows in $W^o$ may not be mapped to any dictionary item, in which case we let $g(s) = 0$. This means such neurons are not close (in angular distance) to any ground-truth after random initialization. Also note that for some rows $W_{s\star}^o$, there might exist multiple $j \in [k]$ such that $g(s) = j$ according to our criterion in the definition. But when $\lambda \leq \frac{1}{2}$, which is always the case by our model assumption on incoherence, Lemma 3 shows that the assignment must be unique, in which case the mapping is well defined.

**Lemma 3** (Uniqueness of firing). *Suppose during training, the weight matrix has a fixed norm $c$. At time $t$, for any row of weight matrix $W_{s\star}^t$, we denote by $\tau_{s,1} := \max_j \langle \frac{W_{s\star}^t}{c}, W_{j\star}^* \rangle$, and we denote by $\tau_{s,2} := \max_{j \in [k], j \neq 1} |\langle \frac{W_{s\star}^t}{c}, W_{j\star}^* \rangle|$. Then for any $\lambda \leq \frac{1}{2}$, $\tau_{s,1} \geq \sqrt{1-\lambda^2} \implies \{\tau_{s,2} < \tau_{s,1}\}$.*

Thus, for any $s \in [n]$ with $g(s) > 0$, the uniqueness of firing condition holds and the mapping $g$ is defined unambiguously. So we simplify notations on measure of distance and closeness as

$$\Delta_s^t := \Delta(W_{s\star}^t, W_{g(s)}^*)$$

$$\phi_s^t := \phi(W_{s\star}^t, W_{g(s)}^*)$$

## 5.2 PART II: THE EVOLVEMENT OF WEIGHTS AND BIAS DURING SGD TRAINING

This section lower bounds the expected increase of $\phi_s^t$ after each SGD update, conditioning on $F^t$. We first show that conditioning on $F^t$, the firing of a neuron $s$ with $g(s) = j$, will indicate that the data indeed comes from the $j$-th component, which is characterized by event $E^t$.

**Definition 3.** *At step $t$, we denote the event of **correct firing** of $W^t$ as*

$$E^t := \{\forall 0 \leq i \leq t, \forall s \ s.t. \ x \sim C_{g(s)}, \langle W_{s\star}^i, x \rangle + b_s^i > 0\}$$

$$\cap \{\forall 0 \leq i \leq t, \forall s \ s.t. \ g(s) > 0 \ and \ x \sim C_j, j \neq g(s), \langle W_{s\star}^i, x \rangle + b_s^i < 0\}$$

---

[3]Note that the firing map $g$ is only defined for the sake of analysis; the algorithm does not have access to this information.

**Definition 4.** *At step $t$, we denote the event of satisfying **local condition** of $W^t$ as*

$$F^t := \{\forall 0 \le i \le t, \forall s \in [n] \ s.t. \ g(s) > 0, \ \langle W_{s\star}^i, W_{g(s)\star}^* \rangle \ge c\sqrt{1 - \lambda^2}\}$$

**Lemma 4** (Correctness of firing). *If at $t \ge 0$, if $\forall j \in [k]$, the network parameters $(W_{s\star}^t, b_s^t)$ is chosen that satisfies*

$$b_s^t = E\langle x 1_{\{a_s^{t-1} > 0\}}, W_{s\star}^t\rangle (\frac{1}{\|W_{s\star}^t\|^2} - 1)$$

*with $\|W_{s\star}^t\| = c$ s.t. such that $\frac{3}{2} \le c \le \sqrt{6k}$. Then for any $t > 0$, $F^t \implies E^t$.*

Then we proceed to characterize the expected change of $\phi_s^t$, conditioning on $E^t$.

**Theorem 2.** *Suppose $E^t$ holds, then after one step of stochastic gradient descent update on $W^t$, $W^{t+1}$ satisfies*

*version 1*

$$E[\phi_s^{t+1} | E^t] \ge \phi_s^t \{1 + \frac{2\eta^t}{k\|W_{s\star}^t\|}(1 - \phi_s^t)\} - (\eta^t)^2 B$$

*version 2*

$$\phi_s^{t+1} \ge \phi_s^t \{1 + \frac{2\eta^t}{k\|W_{s\star}^t\|}(1 - \phi_s^t)\} - \eta^t Z - (\eta^t)^2 B \ \ with \ \ E[Z|E^t] = 0 \ and \ |Z| \le B$$

*for some constant $B > 0$ where $B$ is a constant depending on the model parameter $\|\epsilon\|$ and the norm of rows of weight matrix.*

### 5.3 PART III: CONVERGENCE OF MARTINGALES

By Theorem 2, the sequence $\phi_s^o, \phi_s^1, \dots, \phi_s^t, \dots$ is a sub-martingale. One caveat of is that the expected increase of the cosine of angle between $W_{s\star}^t$ and $W_{g(s)\star}^*$ is conditional on $E^t$, the correct firing condition. So showing that the correct firing event indeed holds is crucial to our overall convergence analysis. Since by Lemma 4, $F^t \implies E^t$, it suffices to show that $F^t$ holds. To this end, note that $F^t$'s form a nested sequence

$$F^o \supset F^1 \supset \dots F^t \supset \dots$$

We denote the limit of this sequence as

$$F^\infty := \lim_{t \to \infty} F^t$$

So $F^\infty$ is the event that

$$\{\langle W_{s\star}^t, W_{j\star}^* \rangle \ge c(1 - \delta_o), \forall t \ge 0, \forall j \in [k] \forall s \in [n] \ s.t. \ g(s) = j\}$$

Theorem 3 shows that $Pr(F^\infty)$ is in fact arbitrarily close to one, conditioning on $F^o$. We note that there is a line of recent work that analyze the convergence of SGD on non-convex functions Balsubramani et al. (2013); Ge et al. (2015); Tang & Monteleoni (2017), where similar technical difficulty arise: to show local improvement of the algorithm on a non-convex functions, one usually needs to lower bound the probability of the algorithm entering a "bad" region, which can be saddle points Ge et al. (2015); Balsubramani et al. (2013) or the part of solution space outside of a local neighborhood Tang & Monteleoni (2017). Some variant of martingale concentration is usually used for obtaining such result. Here, since event $F^t$ can be equivalently interpreted as $W_{s\star}^t$ remains within the local neighborhood of $W_{g(s)\star}^*$ defined as $\{y : \langle y, W_{j\star}^* \rangle \ge c(1 - \delta_o)\}$, we employ a technique similar to that in Tang & Monteleoni (2017) to show that $F^t$ holds with high probability for all $t$.

**Theorem 3.** *Fix any $\delta \in (0, \frac{n}{e})$. Suppose we choose $\eta^t = \frac{c'}{t + t_o}$ such that*

$$c' > 2kc$$

$$t_o \ge 192(c')^2 B^2 (\ln \frac{n}{\delta})^2 (1 + \frac{1}{(\alpha - \lambda)^2})^2$$

*Then conditioning on $F^o$, we have*

$$Pr(F^\infty) = 1 - \delta$$

## 6 OPEN PROBLEMS

There are several interesting questions that are not addressed here. First, as noted in our discussion in Section 2, the update of bias as analyzed in our algorithm is not exactly what is used in original SGD. It would be interesting (and difficult) to explore whether the algorithm has fast convergence when $b^t$ is updated by SGD with a decaying learning rate. Second, our model assumption is rather strong, and it would be interesting to see whether similar results hold on a relaxed model, for example, where one may relax to 1-sparse constraint to $m$-sparse, or one may relax the finite bound requirement on the noise structure. Third, our performance guarantee of random initialization depends on a lower bound on the surface area of spherical caps. Improving this bound can improve the tightness of our initialization guarantee. Finally, it would be very interesting to examine whether similar result holds for activation functions other than ReLu, such as sigmoid function.

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

## 7 Appendix

**Derivation of stochastic gradients** Upon receiving a data point $x$, the stochastic gradient with respect to $W$ is a jacobian matrix whose $(j^*, i^*)$-th entry reads

$$\frac{\partial \ell(W, b; x)}{\partial w_{j^*i^*}} = \frac{\partial}{\partial w_{j^*i^*}} \sum_{i=1}^{d} \frac{1}{2}[x_i - \sum_{j=1}^{n} w_{ji}s(\sum_{l=1}^{d} w_{jl}x_l + b_j)]^2$$

$$= \sum_{i=1}^{d}\{x_i - \sum_{j=1}^{n} w_{ji}s(\sum_{l=1}^{d} w_{jl}x_l + b_j)\}\{-\frac{\partial}{\partial w_{j^*i^*}} \sum_{j=1}^{n} w_{ji}s(\sum_{l=1}^{d} w_{jl}x_l + b_j)\}$$

$$= -\sum_{i=1}^{d}\{x_i - \sum_{j=1}^{n} w_{ji}s(\sum_{l=1}^{d} w_{jl}x_l + b_j)\}\frac{\partial}{\partial w_{j^*i^*}} w_{j^*i}s(\sum_{l=1}^{d} w_{j^*l}x_l + b_{j^*})$$

For $i \neq i^*$, the derivative of the second term can be written using the chain rule as

$$w_{j^*i}\frac{\partial}{\partial w_{j^*i^*}}s(\sum_{l=1}^{d} w_{j^*l}x_l + b_{j^*}) = w_{j^*i}\frac{\partial s(a_{j^*})}{\partial a_{j^*}}x_{i^*}$$

where we let $a_{j^*} := w_{j^*l}x_l + b_{j^*}$, which is the activation of the $j^*$-th neuron upon receiving $x$ in the hidden layer before going through the ReLu unit.

For $i = i^*$, the derivative of the second term can be written using product rule and chain rule as

$$w_{j^*i^*}\frac{\partial}{\partial w_{j^*i^*}}s(\sum_{l=1}^{d} w_{j^*l}x_l + b_{j^*}) + s(\sum_{l=1}^{d} w_{j^*l}x_l + b_{j^*}) = w_{j^*i^*}\frac{\partial s(a_{j^*})}{\partial a_{j^*}}x_{i^*} + s(a_{j^*})$$

Let $r \in \mathbb{R}^d$ be the residual vector with $r_i := x_i - \sum_{j=1}^{n} w_{ji}s(\sum_{l=1}^{d} w_{jl}x_l + b_j)$. Then we have

$$\frac{\partial \ell(W, b; x)}{\partial w_{j^*i^*}} = -\{\sum_{i=1}^{d} r_i w_{j^*i}\frac{\partial s(a_{j^*})}{\partial a_{j^*}}x_{i^*} + s(a_{j^*})r_{i^*}\}$$

In vector notation, the stochastic gradient of loss with respect to the $j$-th row of $W$ can be written as

$$\frac{\partial \ell(W, b; x)}{\partial W_{j\star}} = -(\frac{\partial s(a_j)}{\partial a_j}\langle r, W_{j\star}\rangle x + s(a_j)r) \tag{5}$$

Similarly, we can obtain the stochastic gradient with respect to the $j$-th entry of the bias term as

$$\frac{\partial \ell(W, b; x)}{\partial b_j} = -\frac{\partial s(a_j)}{\partial a_j}\langle r, W_{j\star}\rangle \tag{6}$$

Now let us examine the terms $\frac{\partial s(a_j)}{\partial a_j}$ and $\langle r, W_{j\star}\rangle$. By property of ReLu function,

$$\frac{\partial s(a_j)}{\partial a_j} = \begin{cases} 0 & \text{if } a_j <= 0 \\ 1 & \text{if } a_j > 0 \end{cases}$$

Mathematically speaking, the derivative of ReLu at zero does not exist. Here we follow the convention used in practice by setting the derivative of ReLu at zero to be $0$. In effect, the event $\{a_j = 0\}$ has zero probability, so what derivative to use at zero does not affect our analysis (as long as the derivative is finite).

***Proof of main theorem***. Consider any time $t > 0$. By Lemma 1 and 2, the probability of successfully initializing the network can be lower bounded by

$$Pr(F_*^o) \geq 1 - k\exp\{-n(\frac{\lambda}{\sqrt{2}})^{d-3}\} \quad \text{if the network is initialized with Gaussian variables}$$

or

$$Pr(F_*^o) \geq (1 - k\exp\{-\frac{n\lambda^2}{8kB^2}\})(1 - 3\exp\{-\frac{n^3}{100k^2}\}) \quad \text{if initialized with data}$$

Conditioning on $F_*^o$ and applying Theorem 3, we get that for all $t \geq 0$,

$$Pr(F^t) \geq Pr(F^\infty) \geq 1 - \delta$$

Since $F^t \implies E^t$ by Lemma 4, we can apply version 1 of Theorem 2 to get the expected increase in $\phi_s^t$ for any $s$ such that $g(s) > 0$ as:

$$E[\phi_s^t|F^{t-1}] \geq \phi_s^{t-1}\{1 + \frac{2\eta^{t-1}}{k\|W_{s\star}^{t-1}\|}(1 - \phi_s^{t-1})\} - (\eta^{t-1})^2 B$$

Since by $F^o$, $\forall j \in [k]$, there exists $s \in [n]$ such that $g(s) = j$. Let $s(j)$ be any $s \in [n]$ such that $g(s) = j$. Then, the inequality above translates to

$$E[\Delta_{s(j)}^t|F^{t-1}] \leq \Delta_{s(j)}^{t-1} - \frac{2\eta^{t-1}}{k\|W_{s(j)\star}^t\|}\Delta_{s(j)}^{t-1}(1 - \Delta_{s(j)}^{t-1}) + (\eta^{t-1})^2 B$$

$$= \Delta_{s(j)}^{t-1}(1 - \frac{\beta^{t-1}}{t_o + t - 1}) + \frac{(c')^2}{(t_o + t - 1)^2}B$$

where

$$2 \leq \beta^t = \frac{2c'(1 - \Delta_{s(j)}^t)}{k\|W_{s(j)\star}\|} = \frac{2c'(1 - \Delta_{s(j)}^t)}{kc} \leq \frac{2c'}{kc}$$

by our choice of $c'$ and by our assumption on the initial value $\Delta_{s(j)}^o$. Taking total expectation up to time $t$, conditioning on $F^t$, and letting $\beta$ denote a lower bound on $\beta^t$, we get

$$E[\Delta_{s(j)}^t|F^t] \leq E[\Delta_{s(j)}^{t-1}|F^t](1 - \frac{\beta^{t-1}}{t_o + t - 1}) + \frac{(c')^2}{(t_o + t - 1)^2}B$$

$$\leq E[\Delta_{s(j)}^{t-1}|F^{t-1}](1 - \frac{\beta}{t_o + t - 1}) + \frac{(c')^2}{(t_o + t - 1)^2}B$$

where the last inequality is by the same argument as that in Lemma 8. This has the exact same form as in Lemma D.1 of Balsubramani et al. (2013). Applying it with $u_t := E[\Delta_{s(j)}^t|F_t]$, $a = \beta$, and $b = (c')^2 B$ (note our $t + t_o$ matches their notion of $t$), we get

$$E[\Delta_{s(j)\star}^t|F_t] \leq (\frac{t_o + 1}{t_o + t + 1})^\beta \Delta_{s(j)\star}^o + \frac{(c')^2 B}{\beta - 1}(1 + \frac{1}{t_o + 1})^{\beta+1}\frac{1}{t_o + t + 1}$$

By the upper bound on $\beta^t$, we can choose $\beta$ as small as $\frac{2c'}{kc}$. So we can get an upper expressed in algorithmic and model parameters as

$$E[\Delta_{s(j)\star}^t|F_t] \leq (\frac{t_o + 1}{t_o + t + 1})^{\frac{2c'}{kc}}\Delta_{s(j)\star}^o + \frac{(c')^2 B}{\frac{2c'}{kc} - 1}(1 + \frac{1}{t_o + 1})^{\frac{2c'}{kc}+1}\frac{1}{t_o + t + 1}$$

$$\leq (\frac{t_o + 1}{t_o + t + 1})^{\frac{2c'}{kc}}\frac{\lambda^2}{2} + \frac{(c')^2 B}{\frac{2c'}{kc} - 1}(1 + \frac{1}{t_o + 1})^{\frac{2c'}{kc}+1}\frac{1}{t_o + t + 1}$$

The second inequality holds because by $F^o$,

$$\Delta^o_{s(j)\star} = 1 - \phi^o_{s(j)} \leq 1 - (1 - \frac{\lambda^2}{2}) = \frac{\lambda^2}{2}$$

Finally,

$$E[\Theta(W^t, W^*)|F^t] = \frac{1}{k}\sum_{j\in[k]} E \min_{s\in[n],g(s)=j} \Delta(W^t_{s\star}, W^*_{j\star}) \leq \frac{1}{k}\sum_{j\in[k]} E\Delta(W^t_{s(j)\star}, W^*_{j\star})$$

$$\leq (\frac{t_o+1}{t_o+t+1})^{\frac{2c'}{kc}}\frac{\lambda^2}{2} + \frac{(c')^2 B}{\frac{2c'}{kc}-1}(1+\frac{1}{t_o+1})^{\frac{2c'}{kc}+1}\frac{1}{t_o+t+1}$$

$$\leq (\frac{t_o+1}{t_o+t+1})^4\frac{\lambda^2}{2} + \frac{(c')^2 B}{3}(1+\frac{1}{t_o+1})^{\frac{2c'}{kc}+1}\frac{1}{t_o+t+1}$$

where the last inequality is by our requirement that $c' > 2kc$. $\qquad\square$

## 7.1 PART I: PERFORMANCE GUARANTEE OF INITIALIZATION

***Proof of Lemma 1.*** Let $u = \frac{z}{\|z\|}$, where $z \in \mathbb{R}^d$ with $z_i \sim N(0,1)$. We know that $u$ is a random vector on $S^{d-1}$, the $d$-dimensional unit sphere. For any fixed vector $v \in S^{d-1}$, we have

$$Pr(\langle u, v\rangle \geq 1 - h) = Pr(u \in S_{\text{cap}}(v, h))$$

where $S_{\text{cap}}(v, h)$ is the surface of the spherical cap centered at $v$ with height $h$. By property of spherical Gaussian, we know that $u$ is uniformly distributed on $S^{d-1}$. So we can directly calculate the probability above as

$$Pr(u \in S_{\text{cap}}(v, h)) = \frac{\mu(S_{\text{cap}}(v, h))}{\mu(S^{d-1})}$$

where $\mu$ measures the area of surface. The latter ratio can be lower bounded (see Lemma 5 in the Appendix) as a function of $d$ and $h$:

$$\frac{\mu(S_{\text{cap}}(v, h))}{\mu(S^{d-1})} \geq f(d, h) = \frac{(\sqrt{h(2-h)})^{d-1}}{2d}h$$

Since for all row $i \in [n]$, their entries are $W^o_{i\star} = c\frac{z}{\|z\|}$, then for any ground-truth dictionary item $W^*_{j\star}, j \in [k]$, we have

$$Pr\{\max_{i\in[n]}\langle W^o_{i\star}, v\rangle \geq c(1-h)\} = Pr\{\exists i \in [n], s.t. \langle W^o_{i\star}, v\rangle \geq c(1-h)\}$$

$$= 1 - Pr\{\forall i \in [n], \langle W^o_{i\star}, v\rangle < c(1-h)\}$$

$$= 1 - \Pi^n_{i=1}Pr\{\langle cu, v\rangle < c(1-h)\} = 1 - \Pi^n_{i=1}Pr\{\langle u, v\rangle < 1-h\}$$

$$\geq 1 - \Pi^n_{i=1}(1 - f(d, h)) \geq 1 - \exp^{-nf(d,h)}$$

By union bound, this implies that

$$Pr\{\forall j \in [k], \max_{i\in[n]}\langle W^o_{i\star}, W^*_{j\star}\rangle \geq c(1-h)\} \geq 1 - k\exp^{-nf(d,h)}$$

Now by our choice of the form of lower bound on the inner product, we have

$$h = 1 - \sqrt{1-\rho}$$

substituting this into the function $f(d, h)$, we get a nice form

$$\frac{(\sqrt{\rho})^{d-1}}{1-\sqrt{1-\rho}} = \frac{(\sqrt{\rho})^{d-1}}{1-(1-\rho)}(1+\sqrt{1-\rho}) = (\sqrt{\rho})^{d-3}(1+\sqrt{1-\rho}) \geq (\sqrt{\rho})^{d-3}$$

Substituting this into the previous inequality written in terms of $h$ and letting $\rho = \frac{\lambda^2}{2}$ completes the proof. $\qquad\square$

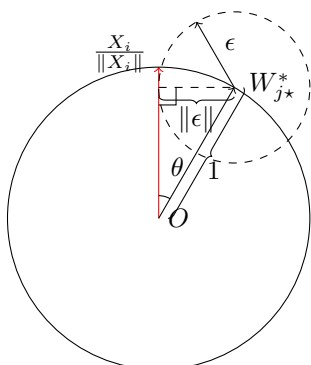

Figure 2: The unit circle lies in $H$. For $\epsilon$ with a fixed norm, the picture illustrates one of the configurations where the maximal angle ($\theta$) between data point $X_i$ (in red) and $W_{j\star}^*$ is achieved.

**Lemma 5** (Lower bound on surface area of spherical cap). *Let $S^{d-1}$ denote the d-dimensional hypersphere, and let $S_{cap}(\nu, h) \subset S^{d-1}$ denote the surface of a hyper spherical cap centered at $\nu$ with height $h$. Let $\mu(\cdot)$ denote measure of area of surface in $d$-dimensional Euclidean space. Let $f(h,d) := (\sqrt{2h - h^2})^{d-1}\frac{h}{2d}$, then*

$$\frac{\mu(S_{cap}(\nu, h))}{\mu(S^{d-1})} \geq f(h,d)$$

*Proof.* Let $B^d$ denote the $d$-dimensional unit ball in Euclidean space, and let $B_{cap}(\nu, h)$ denote the spherical cap centered at $\nu$ with height $h$. Then applying Lemma 4.1 of Micciancio & Voulgaris (2010), we get

$$\frac{Vol(B_{cap})(\nu, h)}{Vol(B^d)} > \sqrt{2h - h^2}^{d-1}\frac{h}{2d}$$

where $Vol(\cdot)$ denotes measure of volume in $\mathbb{R}^d$. So this lower bounds the ratio between **volumes** between spherical cap and the unit ball. We show that we can use this to lower bound the ratio between **surface areas** between spherical cap and the unit ball. Since by **?**, we know that the ratio between their area can be expressed exactly as

$$\frac{\mu(S_{cap}(\nu, h))}{\mu(S^{d-1})} = \frac{1}{2}I_{2h-h^2}(\frac{d-1}{2}, \frac{1}{2})$$

and the ratio between their volume

$$\frac{Vol(B_{cap})(\nu, h)}{Vol(B^d)} = \frac{1}{2}I_{2h-h^2}(\frac{d+1}{2}, \frac{1}{2})$$

where $I_x(a, b)$ is the regularized incomplete beta function. By property of $I_x(a, b)$,

$$I_x(a+1, b) < I_x(a, b)$$

So we have

$$\frac{\mu(S_{cap}(\nu, h))}{\mu(S^{d-1})} > \frac{\mu(B_{cap})(\nu, h)}{\mu(B^d)} > \sqrt{2h - h^2}^{d-1}\frac{h}{2d}$$

$\square$

*Proof of Lemma 2.* For any $X_i \sim C_j$, for any $j \in [k]$. We first claim that

$$\{\|\epsilon_i\|^2 \leq \frac{\lambda}{\sqrt{2}}\} \implies \{\langle\frac{X_i}{\|X_i\|}, W_{j\star}^*\rangle \geq \sqrt{1 - \frac{\lambda^2}{2}}\}$$

*Proof of claim.* Let us consider the two-dimensional plane $H$ determined by $X_i$ and $W_{j\star}^*$. Clearly, $\epsilon_i = W_{j\star}^* - X_i$ also lies in $H$. Let $\theta := \angle(X_i, W_{j\star}^*)$ denote the angle between $X_i$ and $W_{j\star}^*$. Note that $\cos\theta = \langle \frac{X_i}{\|X_i\|}, W_{j\star}^* \rangle$. Fix the norm of noise $\|\epsilon_i\|$. It is clear from elementary geometric insight that $\theta$ is maximized (and hence $\cos\theta$ is minimized when the line of $X_i$ is tangent to the ball centered at $W_{j\star}^*$ with radius $\|\epsilon_i\|$. We can directly calculate the value of $\cos\theta$ at this point (see Figure 2) as $\cos\theta = \sqrt{1 - \|\epsilon_i\|^2}$, which finishes the proof of claim. $\qquad\square$

Now, we denote two events
$$A := \{\min_{j\in[k]} \sum_{i\in[n]} 1_{\{X_i \in C_j\}} \geq \frac{n}{2k}\}$$

and
$$B := \{\max_{j\in[k]} \frac{\sum_{i\in[n]} \|\epsilon_i\|^2 1_{\{X_i \in C_j\}}}{\sum_{i\in[n]} 1_{\{X_i \in C_j\}}} \leq \frac{\lambda}{\sqrt{2}}\}$$

The probability of event $A$ can be lower bounded by concentration inequality for multinomial distribution Devroye (1983): Let $n_j := \sum_{i\in[n]} 1_{\{X_i \in C_j\}}$. We get

$$\Pr(\neg A) = \Pr(\exists j \in [k], n_j - \frac{n}{k} \leq -\frac{n}{2k}) \leq \Pr(\sum_{j=1}^k |n_j - \frac{n}{k}| \geq \frac{n}{2k})$$

$$\leq 3\exp\{-n/25(\frac{n}{2k})^2\} = 3\exp\{-\frac{n^3}{100k^2}\}$$

where the second inequality is by Lemma 3 of Devroye (1983).

$$\Pr(B|A) \geq 1 - k\Pr(\overline{\|\epsilon\|^2} \geq \frac{\lambda}{\sqrt{2}})$$

where $\overline{\|\epsilon\|^2}$ is the empirical mean of $\|\epsilon_i\|^2$ for all $X_i, i \in [n]$ belonging to the same component $C_j$ for some $j \in [k]$. Conditioning on $A$, we know that the average is taken over at least $\frac{n}{2k}$ samples for each component $C_j$. By one-sided Hoeffding's inequality,

$$\Pr(\overline{\|\epsilon\|^2} \geq \frac{\lambda}{\sqrt{2}}) \leq \Pr(\overline{\|\epsilon\|^2} \geq E\|\epsilon\|^2 + \frac{\lambda}{2\sqrt{2}}) \leq \exp\{-\frac{2(n/2k)(\frac{\lambda}{2\sqrt{2}})^2}{B}\} = \exp\{-\frac{n\lambda^2}{8kB}\}$$

(note, we abuse notation $B$ in the exponent as an upper bound on $\|\epsilon\|^2$, as in other parts of the analysis). Thus,

$$\Pr\{F_*^o\} = \Pr\{\min_{j\in[k]} \max_{i\in[n]} \langle \frac{X_i}{\|X_i\|}, W_{j\star}^* \rangle \geq \sqrt{1 - \frac{\lambda^2}{2}}\}$$

$$\geq \Pr\{\max_{j\in[k]} \min_{i\in[n]} \|\epsilon_i\|^2 \leq \frac{\lambda}{\sqrt{2}}\}$$

$$\geq \Pr B \geq \Pr(B \cap A) = \Pr(B|A)\Pr(A) \geq (1 - k\exp\{-\frac{n\lambda^2}{8kB}\})(1 - 3\exp\{-\frac{n^3}{100k^2}\})$$

$\qquad\qquad\qquad\qquad\qquad\qquad\qquad\qquad\qquad\qquad\qquad\qquad\qquad\qquad\qquad\qquad\square$

*Proof of Lemma 3.* To simplify notation, we use $\tau_1$ ($\tau_2$) as a shorthand for $\tau_{s,1}$ ($\tau_{s,2}$). Let $j = \arg\max_{i\in[k]} \langle W_{s\star}^t, W_{i\star}^* \rangle$. Then $\langle W_{s\star}^t, W_{j\star}^* \rangle = c\tau_1$. If $W_{s\star}^t$ and $W_{j\star}^*$ align, then

$$|\langle W_{s\star}^t, W_{j\star}^* \rangle| = c|\langle W_{j\star}^*, W_{j\star}^* \rangle| \leq c\lambda \leq c\tau_{s,1}$$

where the second inequality is by relation of $\lambda$ and $\tau_{s,1}$.

If $W_{s\star}^t$ and $W_{j\star}^*$ do not align, then they determine a two-dimensional plane $H$. Let $i \in [k]$ and $i \neq j$. We can decompose the unit vector $W_{i\star}^*$ as

$$W_{i\star}^* = proj_H(W_{i\star}^*) + orth_H(W_{i\star}^*)$$

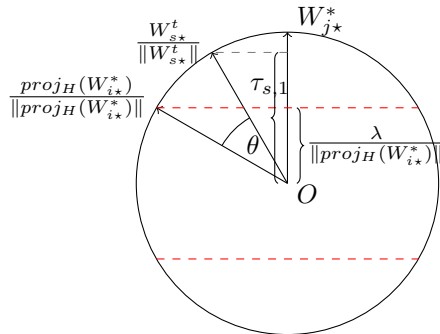

Figure 3: The unit circle lies in $H$. The picture illustrates one of the possible configurations where the minimal angle ($\theta$) between $proj_H(W_{i\star}^*)$ and $W_{s\star}^t$ is achieved.

where $proj_H(W_{i\star}^*)$ is the orthogonal projection of $W_{i\star}^*$ onto $H$. First, note that by orthogonality,

$$|\langle proj_H(W_{i\star}^*), W_{j\star}^* \rangle| \le \lambda$$

Since $proj_H(W_{i\star}^*)$, $W_{s\star}^t$, and $W_{j\star}^*$ lie on the same plane, we can easily see via geometric insight (see Figure 3) that,

$$\langle \frac{proj_H(W_{i\star}^*)}{\|proj_H(W_{i\star}^*)\|}, \frac{W_{s\star}^t}{\|W_{s\star}^t\|} \rangle \le \frac{\lambda}{\|proj_H(W_{i\star}^*)\|}\tau_1 + \sqrt{1 - \frac{\lambda^2}{\|proj_H(W_{i\star}^*)\|^2}}\sqrt{1 - \tau_1^2}$$

Therefore,

$$|\langle W_{i\star}^*, W_{s\star}^t \rangle| \le c(\lambda\tau_1 + \sqrt{\|proj_H(W_{i\star}^*)\|^2 - \lambda^2}\sqrt{1 - \tau_1^2})$$

$$\le c(\lambda\tau_1 + \sqrt{1 - \lambda^2}\sqrt{1 - \tau_1^2})c2\lambda\tau_1$$

Suppose $\tau_1^2 \ge \frac{1+\lambda}{2}$. It can be verified that this implies

$$c\tau_2 = |\langle W_{i\star}^*, W_{s\star}^t \rangle| \le c\sqrt{\frac{1 + \lambda}{2}} = c\tau_1$$

Observing that $\tau_1^2 \ge \frac{1+\lambda}{2}$ indeed holds provided $\lambda \le \frac{1}{2}$ and $\tau_1 \ge \sqrt{1 - \lambda^2}$ completes the proof. □

### 7.2 PART II: THE EVOLVEMENT OF WEIGHTS AND BIAS DURING SGD TRAINING

**Proof of Lemma 4.** Conditioning on $F^t$, we show that $E^t$ holds by induction on $0 \le i \le t$. We start by showing that $E^o$ holds.

**Base case** In this case

$$b_s^o = E\langle x, W_{s\star}^o \rangle (\frac{1}{c^2} - 1)$$

Note that by our model assumption,

$$E\langle x, W_{s\star}^o \rangle = \sum_{i \in [k]} \frac{1}{k} \langle W_{s\star}^o, W_{i\star}^* \rangle$$

Since $F^o$ holds, we know that the firing of neuron $s$ is unique. Let $\tau_{s,1}$ and $\tau_{s,2}$ as defined in Lemma 3.

Let $x \sim C_j$. Consider the case $g(s) = j$. In this case, Lemma 3 implies that

$$\tau_{s,1} := \max_{j'} \langle W_{s\star}^o, W_{j'\star}^* \rangle = \langle W_{s\star}^o, W_{j\star}^* \rangle$$

We will repeatedly use the following relation, as proven by Lemma 6,

$$\tau_{s,2} := \max_{i \neq j} |\langle W_{s\star}^o, W_{i\star}^* \rangle|$$

We have

$$W_{s\star}^o x + b_s^o = (W_{s\star}^o)^T (W_{j\star}^* + \epsilon) + (\frac{1}{c^2} - 1) \sum_{i \in [k]} \frac{1}{k} \langle W_{s\star}^o, W_{i\star}^* \rangle$$

Now, observe that $\frac{1}{c^2} - 1 < 0$ and

$$\sum_{i \in [k]} \frac{1}{k} \langle W_{s\star}^o, W_{i\star}^* \rangle \le c\frac{1}{k}\tau_{s,1} + c\frac{k-1}{k}\tau_{s,2}$$

So we get,

$$W_{s\star}^o x + b_s^o \ge c\tau_{s,1} - c\|\epsilon\| + (\frac{1}{c^2} - 1)(c\frac{1}{k}\tau_{s,1} + c\frac{k-1}{k}\tau_{s,2})$$

$$\ge c\tau_{s,1} - c\|\epsilon\| + (\frac{1}{c^2} - 1)(c\frac{1}{k}\tau_{s,1} + c\frac{k-1}{k}c2\lambda\tau_{s,1})$$

$$= c\{\tau_{s,1}\{1 - (1 - \frac{1}{c^2})(\frac{1}{k} + 2\lambda)\} - \|\epsilon\|\}$$

Since by our assumption, $\lambda \le \frac{1}{8k}$, $(1 - \frac{1}{c^2})(\frac{1}{k} + 2\lambda) \le (1 - \frac{1}{c^2})\frac{5}{4k} \le \frac{5}{4k}$. Furthermore, since by our assumption, $\|\epsilon\| \le \frac{\sqrt{1-\lambda^2}}{6k} \le \sqrt{1-\lambda^2}(1 - \frac{5}{4k}) \le (1 - \frac{5}{4k})\tau_1$ we get $W_{s\star}^o x + b_s^o \ge 0$.

Consider the case $g(s) = j'$, $j' \ne j$. We first upper bound $b_o$ in this case. Since $\frac{1}{c^2} - 1 < 0$, we would like to lower bound

$$E\langle x, W_{s\star}^o \rangle = \sum_{i \in [k]} \frac{1}{k} \langle W_{s\star}^o, W_{i\star}^* \rangle = \frac{1}{k}\left\{ \langle W_{s\star}^o, W_{j'\star}^* \rangle + \sum_{i \ne j} \langle W_{s\star}^o, W_{i\star}^* \rangle \right\}$$

$$\ge \frac{1}{k}\left\{ \langle W_{s\star}^o, W_{j'\star}^* \rangle - |\sum_{i \ne j} \langle W_{s\star}^o, W_{i\star}^* \rangle| \right\} \ge \frac{1}{k}c\tau_{s,1} - \frac{k-1}{k}c\tau_{s,2} \ge \frac{1}{k}c\tau_{s,1} - c2\lambda\tau_{s,1}$$

On the other hand,

$$W_{s\star}^o x \le |\langle W_{s\star}^o, W_{j\star}^* + \epsilon \rangle| \le c(\tau_{s,2} + \|\epsilon\|)$$

So

$$W_{s\star}^o x + b_s^o \le c(\tau_{s,2} + \|\epsilon\|) - (1 - \frac{1}{c^2})c\tau_{s,1}(\frac{1}{k} - 2\lambda)$$

$$\le c(2\lambda\tau_{s,1} + \|\epsilon\|) - (1 - \frac{1}{c^2})c\tau_{s,1}(\frac{1}{k} - 2\lambda) \le 0$$

where the last inequality is by assumptions $c > \frac{3}{2}$ and $\|\epsilon\| \le \frac{\sqrt{1-\lambda^2}}{6k}$.

**Case** $0 < i \le t$  Suppose $E^i$ holds, we show that $E^{i+1}$ holds for $i \le t - 1$. Let $x \sim C_j$ for any $j \in [k]$. Since $E^i$ holds, we know that

$$\forall s \text{ s.t. } g(s) > 0, a_s^i > 0 \text{ iff } g(s) = j$$

So for neurons $W_{s\star}^{i+1}$ with $g(s) > 0$ and $g(s) \ne j$, we know that these neurons are not updated, that is,

$$W_{s\star}^{i+1} = W_{s\star}^i \text{ and } b_s^{i+1} = b_s^i$$

This means the two conditions in $E^{i+1}$ on neurons $g(s) = j' \ne j$.
For neurons $W_{s\star}^{i+1}$ with $g(s) = j$, $b_s^{i+1}$ is set such that

$$b_s^{i+1} = E\langle x1_{\{a_s^i > 0\}}, W_{s\star}^{i+1}\rangle(\frac{1}{c^2} - 1) = \langle E[x|x \sim C_j], W_{s\star}^{i+1}\rangle(\frac{1}{c^2} - 1) = \langle W_{s\star}^{i+1}, W_{j\star}^*\rangle(\frac{1}{c^2} - 1)$$

Consider the case $x' \sim C_j$, we have

$$(W_{s\star}^{i+1})^T x' + b_s^{i+1} = \epsilon^T W_{s\star}^{i+1} + \frac{1}{c^2}(W_{s\star}^{i+1})^T W_{j\star}^* \ge \tau_{s,1}\frac{1}{c} - c\|\epsilon\| \ge \sqrt{1-\lambda^2}\frac{1}{c} - c\|\epsilon\|$$

where the last inequality is by assumption $\tau_{s,1}^2 \geq 1 - \lambda^2$. It follows that $(W_{s\star}^{i+1})^T x' + b_s^{i+1} > 0$ holds by our assumptions that $c \leq \sqrt{6k}$, and that $\|\epsilon\| \leq \frac{\sqrt{1-\lambda^2}}{6k}$.

Finally, consider the case $x' \sim C_{j'}$ for $j' \neq j, j' \in [k]$. Again, since $F^{i+1}$ holds,

$$\tau_{s,1} = (W_{s\star}^{i+1})^T W_{j\star}^* \geq c\sqrt{1-\lambda^2}$$

we can apply Lemma 3 to get

$$(W_{s\star}^{i+1})^T x' + b_s^{i+1} \leq c\tau_{s,2} + \epsilon^T W_{s\star}^{i+1} + (W_{s\star}^{i+1})^T W_{j\star}^*(\frac{1}{c^2} - 1) \leq c\tau_{s,2} + c\|\epsilon\| + c\tau_{s,1}(\frac{1}{c^2} - 1) \leq 0$$

where the last inequality holds similarly as in the base case. $\square$

**Lemma 6.** *Let $\tau_{s,1}$, $\tau_{s,2}$ be as defined in Lemma 3, and let $\lambda$ be the incoherence parameter. If $\tau_{s,1}^2 \geq 1 - \lambda^2$, then $\tau_{s,2} \leq c\lambda\tau_{s,1}$.*

*Proof.* Using the same argument as the proof of Lemma 3, we get

$$|\langle W_{i\star}^*, W_{s\star}^t \rangle| \leq c(\lambda\tau_1 + \sqrt{\|proj_H(W_{i\star}^*)\|^2 - \lambda^2}\sqrt{1 - \tau_1^2})$$

$$\leq c(\lambda\tau_1 + \sqrt{1-\lambda^2}\sqrt{1-\tau_1^2})$$

$$\leq c2\lambda\tau_1$$

where the last inequality is by the relation between $\tau_1$ and $\lambda$. $\square$

***Proof of Theorem 2.*** We start by proving version 1. The proof of version 2 follows directly. Let $x \sim C_j$ for any $j \in [k]$. We consider and any neuron $s$ s.t. $g(s) = j$.
Again, since $E^t$ holds,

$$a_s^t > 0 | x \sim C_j, g(s) = j$$

We now examine $E[\phi_s^{t+1} | x \sim C_j, g(s) = j, E^t]$. To ease notation, we let $w := W_{s\star}^t$, $w^* := W_{j\star}^*$, $\eta := \eta_w^t$, $r := r^t$, $a := a_s^t$, and $b := b_s^t$ in the proof.

$$\phi_s^{t+1} = \frac{(\langle W_{j\star}^{t+1}, w^* \rangle)^2}{\|W_{j\star}^{t+1}\|^2} = \frac{((w + \eta[\langle r, w \rangle x + ar])^T w^*)^2}{\|w + \eta[\langle r, w \rangle x + ar]\|^2}$$

The denominator reads

$$\|w\|^2 + 2\eta[(r^T w)(x^T w) + a(r^T w)] + \eta^2[(r^t w)^2 \|x\|^2 + 2a(r^T w)(x^T r) + a^2 \|r\|^2]$$

$$= \|w\|^2(1 + \frac{1}{\|w\|^2} 2\eta[(r^T w)(x^T w) + a(r^T w)] + \frac{1}{\|w\|^2} \eta^2[(r^T w)^2 \|x\|^2 + 2a(r^T w)(x^T r) + a^2 \|r\|^2])$$

Let $A(w) := (r^T w)(x^T w) + a(r^T w)$ and $B(w) := (r^T w)^2 \|x\|^2 + 2a(r^T w)(x^T r) + a^2 \|r\|^2$.

$$\frac{1}{\|w + \eta[\langle r, w \rangle x + ar]\|^2} = \frac{1}{\|w\|^2(1 + \frac{1}{\|w\|^2} 2\eta A(w) + \frac{1}{\|w\|^2} \eta^2 B(w))}$$

$$= \frac{1 - (\frac{1}{\|w\|^2} 2\eta A(w) + \frac{1}{\|w\|^2} \eta^2 B(w))}{\|w\|^2\{1 - (\frac{1}{\|w\|^2} 2\eta A(w) + \frac{1}{\|w\|^2} \eta^2 B(w))^2\}}$$

$$\geq \frac{1 - (\frac{1}{\|w\|^2} 2\eta A(w) + \frac{1}{\|w\|^2} \eta^2 B(w))}{\|w\|^2}$$

Let $C(w^*) := (r^T w)(x^T w^*) + a(r^T w^*)$. The numerator reads

$$(w^T w^* + \eta[(r^T w)(x^T w^*) + a(r^T w^*)])^2 = (w^T w^*)^2 + 2\eta(w^T w^*)C(w^*) + \eta^2 C(w^*)^2$$

Putting these together,

$$\phi_j^{t+1} \geq \frac{1}{\|w\|^2}\{(w^Tw^*)^2 + 2\eta(w^Tw^*)C(w^*) + \eta^2 C(w^*)^2\}\{1 - (\frac{1}{\|w\|^2}2\eta A(w) + \frac{1}{\|w\|^2}\eta^2 B(w))\}$$

$$= \frac{1}{\|w\|^2}\{(w^Tw^*)^2 + 2\eta(w^Tw^*)C(w^*) - \frac{1}{\|w\|^2}2\eta A(w)(w^Tw^*)^2\} \quad (7)$$

$$+\frac{1}{\|w\|^2}\{\eta^2 C(w^*)^2 - 4\eta^2 \frac{A(w)C(w^*)(w^Tw^*)^3}{\|w\|^2} - 2\eta^3 \frac{A(w)C(w^*)}{\|w\|^2} - \eta^2 \frac{(w^Tw^*)^2 B(w)}{\|w\|^2}$$

$$-2\eta^3 \frac{B(w)C(w^*)(w^Tw^*)}{\|w\|^2} - \eta^4 \frac{B(w)C(w^*)}{\|w\|^2}\}$$

**Bounding major direction of improvement**   Now let us focus on line (7), which can be viewed as the major direction of change guided by the stochastic gradient, while the rest terms can be viewed as noise. It equals

$$\phi_j^t + 2\eta \frac{(w^Tw^*)C(w^*)}{\|w\|^2} - \frac{1}{\|w\|^2}\phi_j^t 2\eta A(w) = \phi_j^t + \frac{2\eta}{\|w\|^2}\{(w^Tw^*)C(w^*) - \phi_j^t A(w)\}$$

Note that both terms $C(w^*)$ and $A(w)$ are stochastic, since their value is determined by data we sampled from the model distribution. We continue with the key step of our analysis: bounding $EC(w^*)$ and $EA(w)$. Since

$$r^Tw = x^Tw - a\|w\|^2 \quad \text{and} \quad a = x^Tw + b$$

$$(r^Tw)(x^Tw^*) = (x^Tw - a\|w\|^2)(x^Tw^*) = (x^Tw - (x^Tw + b)\|w\|^2)(x^Tw^*)$$
$$= w^Txx^Tw^* - (x^Tw + b)\|w\|^2(x^Tw^*)$$

and

$$a(r^Tw^*) = (x^Tw + b)(x^Tw^* - a(w^Tw^*))$$

So

$$C(w^*) = (2 - \|w\|^2)(w^Txx^Tw^*) - a(x^Tw)(w^Tw^*) - b\|w\|^2(x^Tw^*) + b(x^Tw^*) - ab(w^Tw^*)$$
$$= (2 - \|w\|^2)(w^Txx^Tw^*) - (x^Tw)^2(w^Tw^*)$$
$$-b\|w\|^2(x^Tw^*) + b(x^Tw^*) - 2b(x^Tw)(w^Tw^*) - b^2(w^Tw^*)$$

Since $b$ is chosen such that $b = (w^Tw^*)(\frac{1}{\|w\|^2} - 1)$ conditioning on $E^t$, after some calculation we get

$$E_x\{-2b(x^Tw)(w^Tw^*) - b\|w\|^2(x^Tw^*) + b(x^Tw^*)|\Omega_t\}$$
$$= 2(w^Tw^*)^3(1 - \frac{1}{\|w\|^2}) + (w^Tw^*)(1 - \frac{1}{\|w\|^2})(\|w\|^2 - 1)$$

On the other hand, note that

$$E_x[w^Txx^Tw^*|\Omega_t] = w^TExx^Tw^* = w^TE(w^* + \epsilon)(w^* + \epsilon)^Tw^* = w^T(w^*(w^*)^T + E\epsilon\epsilon^T)w^*$$
$$= w^T(w^*(w^*)^T + \sigma I)w^* = w^Tw^* + \sigma w^Tw^*$$

where the second inequality is by the condition that $x$ is drawn from the $j$-th component, and fourth inequality is due to our model assumption on the covariance structure of $\epsilon$. Similarly,

$$E_x[(x^Tw)^2|\Omega_t] = E_x[w^Txx^Tw|\Omega_t] = w^T(w^*(w^*)^T + \sigma I)w = (w^Tw^*)^2 + \sigma\|w\|^2$$

Therefore, we can calculate the other part as

$$E_x\{(2 - \|w\|^2)(w^Txx^Tw^*) - (x^Tw)^2(w^Tw^*)|\Omega_t\}$$
$$= (2 - \|w\|^2)(w^Tw^*)(1 + \sigma) - [(w^Tw^*)^3 + \sigma(w^Tw^*)\|w\|^2]$$

Combining, we get

$$E_x\{C(w^*)|\Omega_t\} = (2 - \|w\|^2)(w^Tw^*)(1 + \sigma) - [(w^Tw^*)^3 + \sigma(w^Tw^*)\|w\|^2]$$
$$+2(w^Tw^*)^3(1 - \frac{1}{\|w\|^2}) + (w^Tw^*)(1 - \frac{1}{\|w\|^2})(\|w\|^2 - 1) - E_x[b^2(w^Tw^*)|\Omega_t]$$

Thus,

$$E_x\{C(w^*)|\Omega_t\}\frac{(w^T w^*)}{\|w\|^2} = \phi_j^t\{(w^T w^*)^2[1 - \frac{1}{\|w\|^2}] + 2\sigma - 2\sigma\|w\|^2 + \frac{1}{\|w\|^2}\} - E_x[b^2|\Omega_t]\phi_j^t$$

Now, we turn to the term $A(w)$, which equals

$$A(w) = ((x - aw)^T w)(x^T w + a) = [x^T w - (x^T w + b)\|w\|^2](2x^T w + b)$$
$$= (2 - 2\|w\|^2)(x^T w)^2 + b(x^T w)(1 - 3\|w\|^2) - b^2\|w\|^2$$

So

$$E_x[A(w)|\Omega_t] = (2 - 2\|w\|^2)[(w^T w^*)^2 + \sigma\|w\|^2]$$
$$+ (w^T w^*)^2(1 - 3\|w\|^2)(1 - \frac{1}{\|w\|^2}) - E_x[b^2\|w\|^2|\Omega_t]$$

This implies that

$$E_x[A(w)|\Omega_t]\frac{\phi_j^t}{\|w\|^2} = (\phi_j^t)^2(\frac{1}{\|w\|^2} - 2 + \|w\|^2) + \sigma\phi_j^t(2 - 2\|w\|^2) - E_x[b^2|\Omega_t]\phi_j^t$$

Finally, combining the derivation for expected values relevant to $C(w^*)$ and $A(w)$, we get

$$E_x\{C(w^*)|\Omega_t\}\frac{(w^T w^*)}{\|w\|^2} - E_x[A(w)|\Omega_t]\frac{\phi_j^t}{\|w\|^2} = \frac{1}{\|w\|^2}\phi_j^t(1 - \phi_j^t)$$

So the expectation of line (7) is

$$\phi_s^t + 2\eta\frac{1}{\|w\|^2}\phi_s^t(1 - \phi_s^t)$$

Also, note that since the terms $w, w^*, x$ are all bounded, the noise term must be bounded by some constant $B_1$. Therefore, we can get a lower bound on increase of conditional expectation

$$E[\phi_s^{t+1}|x \sim C_j, g(s) = j, E^t] \geq \phi_s^t + 2\eta\frac{1}{\|w\|^2}\phi_s^t(1 - \phi_s^t) - \eta^2 B_1$$

Since $Pr(x \sim C_j) = \frac{1}{k}$, we get

$$E[\phi_s^{t+1}|g(s) = j, E^t] \geq \frac{1}{k}\{\phi_s^t + 2\eta\frac{1}{\|w\|^2}\phi_s^t(1 - \phi_s^t) - \eta^2 B_1\}$$
$$+ \frac{k-1}{k}E[\phi_s^{t+1}|g(s) = j, E^t, x \sim C_{j'}, j' \neq j]$$
$$\geq \phi_s^t + 2\eta\frac{1}{k\|w\|^2}\phi_s^t(1 - \phi_s^t) - \eta^2 B_1$$

where the last inequality holds because $a_s^t(x) \leq 0$ for $x \sim C_{j'}$ and $g(s) = j$ ($j \neq j'$) by $E^t$, and for neuron $s$ such that $a_s^t(x) \leq 0$, the gradient evaluates to zero so those entries are not updated after $t$. Thus, since $E^t$ holds, we know that

$$W_{s\star}^{t+1} = W_{s\star}^t | g(s) > 0, g(s) \neq j, \text{ and } x \sim C_j$$

**version 2** Let
$$Y := \frac{1}{\|w\|^2}(w^T w^*)C(w^*) - \frac{1}{\|w\|^4}A(w)(w^T w^*)^2$$

Then, from the proof of version 1, we know that

$$\phi_s^{t+1} \geq \phi_s^t + 2\eta^t E[Y|E^t, X \sim C_{g(s)}]Pr(X \sim C_{g(s)}) - 2\eta^t E[Y|E^t] + 2\eta^t Y + (\eta^t)^2 B_1$$
$$= \phi_s^t + 2\eta^t \frac{1}{\|w\|^2}\phi_s^t(1 - \phi_s^t) + 2\eta^t(Y - E[Y|E^t]) + (\eta^t)^2 B_1$$

So letting $Z := Y - E[Y|E^t]$, we know that $E[Z|E^t] = 0$ Again, since the terms $w, w^*, x$ are all bounded, there must exists $B_2 \geq |Z|$. Letting $B := \max\{B_1, B_2\}$ finishes the proof. $\square$

### 7.3 PART III: CONVERGENCE OF MARTINGALES

***Proof outline of Theorem 3.*** Conditioning on $F^o$, note that $F^\infty$ can be written as the union of events

$$\cup_{s\in[n],g(s)>0}\{\langle W_{s\star}^t, W_{j\star}^*\rangle \geq c\sqrt{1-\lambda^2}, \forall t\geq 0, \forall j\in[k] \text{ s.t. } g(s)=j\}$$

We denote these individual events regarding neuron $s$ by $F_s^\infty$, that is,

$$F_s^\infty := \{\langle W_{s\star}^t, W_{j\star}^*\rangle \geq c\sqrt{1-\lambda^2}, \forall t\geq 0, \forall j\in[k] \text{ s.t. } g(s)=j\}$$

The proof strategy is to show that the individual probability can be lower bounded by $\frac{\delta}{n}$, and taking the union bound over all neurons. Since $(F_s^\infty)^c$ can be partitioned into a union of disjoint subsets $\cup_{t=0}^\infty (F_s^t \setminus F_s^{t+1})$, where

$$F_s^t := \{\langle W_{s\star}^t, W_{j\star}^*\rangle \geq c\sqrt{1-\lambda^2}, \forall 0\leq i\leq t, \forall j\in[k] \text{ s.t. } g(s)=j\}$$

We can define $E_s^t$ similarly, and note we can easily show that

$$F_s^t \implies E_s^t$$

by going through Lemma 4 exactly the same. To lower bound $Pr(F_s^\infty)$, we upper bound the individual error probability $Pr(F_s^t \setminus F_s^{t+1})$, and show that their summation vanishes for $t\to\infty$. The same approach is taken in Proposition 2 Tang & Monteleoni (2017), and our main idea is to adapt the analysis there to our case. Thanks to the special form of inequality obtained in Theorem 2, we can neatly re-write the statement of its version 2 in terms of $\Delta_s^t$ as

$$\Delta_s^{t+1} \leq \Delta_s^t\{1 - \frac{2\eta^t}{k\|W_{s\star}^t\|}(1-\Delta_s^t)\} + 2\eta^t Z + (\eta^t)^2 B \tag{8}$$

Essentially, the sufficient condition for the analysis of Proposition 2 Tang & Monteleoni (2017) to work are

- The expected decrease on the objective function $\Delta_s^t$ is of the form

$$\Delta_s^{t+1} \leq \Delta_s^t\{1 - \frac{\beta^t}{t+t_o}\} + 2\frac{c'}{t+t_o}Z + (\frac{c'}{t+t_o})^2 B \tag{9}$$

- $\beta^t \geq 2, \ \forall t\geq 0$
- The noise terms $Z, B$ are bounded, and $E[Z|F^t] = 0$.

By our choice of $\eta^t$, the relation in (8) satisfies the special form in (9). Since conditioning on $F^t$, we have

$$\beta^t := \frac{2c'(1-\Delta_s^t)}{k\|W_{s\star}^t\|} \geq \frac{2c'\phi_s^o}{kc}$$

and since conditioning on $F^o$,

$$\phi_s^o \geq \frac{1+(1-\delta_o)^2}{2} \geq \frac{1}{2}$$

for $g(s) > 0$ and our choice of $\delta_o$ in $F^o$. Since we also set

$$c' \geq 2kc$$

we get that $\beta^t \geq 2$ always hold. On the other hand, here the noise terms are obviously bounded by our model assumption. Therefore, we only need to slightly adapt Proposition 2 Tang & Monteleoni (2017) for our purpose. The exact proof is included for completeness. □

***Complete proof of Theorem 3.*** We consider each $F_s^t$ individually. Since the proof for each $s$ is exactly the same, we abuse the notation $F^t$ to let it denote $F_s^t$ for any fixed $s$. Similarly, we let $\Delta^t$ denote $\Delta_s^t$. Conditioning on $F^t$, we know that $\exists \beta > 2$ s.t. $\beta \leq \beta^t$. By Lemma 7, for any $\lambda > 0$, and any $0\leq i\leq t-1$, we have

$$E\{e^{\lambda\Delta^{i+1}}|F^i\} \leq E\{e^{\lambda\{(1-\frac{\beta^i}{t_o+i})\Delta^i}|F^i\} \exp\{\frac{\lambda(c')^2 B}{(t_o+i)^2} + \frac{\lambda^2(c')^2 B^2}{2(t_o+i)^2}\}$$

$$\leq E\{e^{\lambda^{(1)}\Delta^i}|F^{i-1}\} \exp\{\frac{\lambda(c')^2 B}{(t_o+i)^2} + \frac{\lambda^2(c')^2 B^2}{2(t_o+i)^2}\}$$

where $\lambda^{(1)} = \lambda(1 - \frac{\beta}{t_o+i})$, and the second inequality is by Lemma 8. For $k \geq 1$, we define

$$\lambda^{(0)} := \lambda \quad \text{and} \quad \lambda^{(k)} := \Pi_{t=1}^{k}(1 - \frac{\beta}{t_o + (i - t + 1)})\lambda^{(0)}$$

We can similarly get, for $k = 0, \ldots, i$,

$$E\{e^{\lambda^{(k)}\Delta^{i-k+1}}|F^{i-k}\} \leq E\{e^{\lambda^{(k+1)}\Delta^{i-k}}|F^{i-(k+1)}\}$$
$$\exp\{\frac{\lambda^{(k)}(c')^2 B}{(t_o + i - k)^2} + \frac{(\lambda^{(k)})^2(c')^2 B^2}{2(t_o + i - k)^2}\}$$

Since $\forall \beta > 0, k \geq 1$,

$$\lambda^{(k)} = \lambda\Pi_{t=1}^{k}(1 - \frac{\beta}{t_o + (i - t + 1)}) \leq \lambda(\frac{t_o + i - k + 1}{t_o + i})^\beta$$

Since the bound is shrinking as $\beta$ increases and $\beta \geq 2$,

$$\frac{\lambda^{(k)}}{(t_0 + i - k)^2} \leq (\frac{t_o + i - k + 1}{t_o + i})^2 \frac{\lambda}{(t_o + i - k)^2} \leq \frac{4\lambda}{(t_o + i)^2}$$

Recursively applying the relation until we get to the term

$$E\{e^{\lambda^{(i)}\Delta^1}|F^o\} \leq E\{e^{\lambda^{(i+1)}\Delta^o}|F^o\} \exp\{\frac{\lambda(c')^2 B}{(t_o + i)^2} + \frac{\lambda^2(c')^2 B^2}{2(t_o + i)^2}\}$$
$$= \exp\{\lambda^{(i+1)}\Delta^o\} \exp\{\frac{\lambda(c')^2 B}{(t_o + i)^2} + \frac{\lambda^2(c')^2 B^2}{2(t_o + i)^2}\}$$

Combining all these recursive inequalities with the bound on $\lambda^{(k)}$, we get

$$E\{e^{\lambda\Delta^{i+1}}|F^i\} \leq e^{\lambda^{(i+1)}\Delta^0} \exp\{\sum_{k=0}^{i-1}(\frac{4\lambda(c')^2 B}{(t_o + i)^2} + \frac{4\lambda^2(c')^2 B^2}{2(t_o + i)^2})\}$$
$$\leq \exp\{\lambda(\frac{t_o}{t_o + i})^\beta \Delta^0 + [\lambda(c')^2 B + \frac{\lambda^2(c')^2 B^2}{2}]\frac{4i}{(t_o + i)^2}\}$$

Let $\tau_o := \lambda^2$. Then we can apply the conditional Markov's inequality, for any $\lambda_i > 0$,

$$Pr(F^i \setminus F^{i+1}) = Pr(\Delta^{i+1} > \tau_o|F^i)$$
$$= Pr(e^{\lambda_i\Delta^{i+1}} > e^{\lambda_i\tau_o}|F^i) \leq \frac{E[e^{\lambda_i\Delta^{i+1}}|F^i]}{e^{\lambda_i\tau_o}}$$

Since event $F^o = \{\phi^o \geq 1 - \frac{\lambda^2}{2}\}$ implies $\Delta^o = 1 - \phi^o \leq \frac{\lambda^2}{2} = \frac{\tau_o}{2}$, we have

$$(\frac{t_o}{t_o + i})^\beta \Delta^o - \tau_o \leq \Delta^o - \tau_o \leq \frac{\tau_o}{2}$$

Combining this with the upper bound on $E[e^{\lambda_i\Delta^{i+1}}|F^i]$, we get

$$Pr(F^i \setminus F^{i+1}) \leq \exp\left\{-\lambda_i\{\frac{\tau_o}{2} - (B + \frac{\lambda_i B^2}{2})\frac{4(c')^2 i}{(t_o + i)^2}\}\right\}$$

We choose $\lambda_i = \frac{1}{\Delta}\ln\frac{(i+1)^2}{\delta}$ with $\Delta = \frac{\tau_o}{4}$, and show that $\frac{\tau_o}{2} - (B + \frac{\lambda_i B^2}{2})\frac{4(c')^2 i}{(t_o+i)^2}$ is lower bounded by $\Delta$.

**Case 1:** $B > \frac{\lambda_i B^2}{2}$. Since $t_o \geq \frac{32(c')^2 B}{\tau_o}$, we get

$$\frac{1}{2}\tau_o - (B + \frac{\lambda_i B^2}{2})\frac{4(c')^2 i}{(t_o + i)^2} \geq \Delta$$

**Case 2:** $B \leq \frac{\lambda_i B^2}{2}$. We get

$$\frac{1}{2}\tau_o - (B + \frac{\lambda_i B^2}{2})\frac{4(c')^2 i}{(t_o + i)^2}$$

$$\geq 2\Delta - \lambda_i B^2 \frac{4(c')^2 i}{(t_o + i)^2}$$

$$= 2\Delta - \frac{1}{\Delta}\ln\frac{(1+i)^2}{\delta}\frac{4(c')^2 B^2 i}{(t_o + i)^2}$$

$$\geq 2\Delta - \frac{1}{\Delta}\ln\frac{(t_o + i)^2}{\delta}\frac{4(c')^2 B^2 (t_o + i)}{(t_o + i)^2}$$

Now we show

$$\frac{1}{\Delta}\ln\frac{(t_o + i)^2}{\delta}\frac{4(c')^2 B^2}{t_o + i} \leq \Delta$$

Since

$$t_o + i \geq t_o \geq \frac{192(c')^2 B^2}{\tau_o^2}\ln^2\frac{1}{\delta}$$

$\ln\frac{1}{\delta} \geq 1$, and obviously, $\frac{16(c')^2 B^2}{\Delta^2} \geq \frac{16(c')^2 B^2}{(\frac{1}{2}\tau_o)^2} \geq \frac{1}{3}$, we can apply Lemma **??** with $b = 2$, $C :=$ $\frac{16(c')^2 B^2}{(\frac{1}{2}\tau_o)^2}$, $t := t_o + i \geq (\frac{3C}{b-1}\ln\frac{1}{\delta})^{\frac{2}{b-1}}$, and get

$$\frac{4(c')^2 B^2}{\Delta^2}\ln\frac{(t_o + i)^2}{\delta} := 2C\ln t + C\ln\frac{1}{\delta} < t^{b-1} = t_o + i$$

Or equivalently, $\frac{1}{\Delta}\ln\frac{(t_o+i)^2}{\delta}\frac{4(c')^2 B^2}{t_o+i} \leq \Delta$. Thus, for both cases,

$$2\Delta - (B + \frac{\lambda_i B^2}{2})\frac{4(c')^2 i}{(t_o + i)^2} \geq= \Delta$$

This implies

$$Pr(F^i \setminus F^{i+1}) \leq e^{-\frac{1}{\Delta}(\ln\frac{(1+i)^2}{\delta})\Delta} = \frac{\delta}{(i+1)^2}$$

Finally, we have

$$Pr(\cup_{i\geq 1}F^i \setminus F^{i+1}) \leq \sum_{i=1}^{\infty} Pr(F^i \setminus F^{i+1}) \leq \delta$$

Now recall that this holds for each $s$, that is, $\forall s \in [n]$,

$$Pr(F_s^{\infty}) \leq \delta_s$$

substituting $\delta_s = \frac{\delta}{k}$ for $\delta$ in the proof above, and taking the union bound completes the proof. $\square$

**Lemma 7** (Inequality of moment generating function). *Suppose the conditions of Theorem 2 hold. Then conditioning on $E^t$ ($F^t$), we can upper bound the moment generating function of $\Delta^{t+1}$ as*

$$E[e^{\lambda\Delta^{t+1}}|F^t] \leq \exp\{\lambda\Delta^t(1 - \frac{\beta^t}{t + t_o}) + \lambda(\eta^t)^2 B + \lambda^2\frac{(\eta^t)^2 B^2}{2}\}$$

*Proof.* We apply the result of Theorem 2. Rewriting version 2 of Theorem 2 using $\Delta^t := 1 - \phi_s^t$, for any $s \in [n]$, we get

$$\Delta^{t+1} \leq \Delta^t - \frac{2\eta^t}{k\|W_{s\star}^t\|}\Delta^t(1 - \Delta^t) + 2\eta^t Z + (\eta^t)^2 B$$

$$= \Delta^t\{1 - \frac{2\eta^t}{k\|W_{s\star}^t\|}(1 - \Delta^t)\} + 2\eta^t Z + (\eta^t)^2 B$$

We let $\beta^t := \frac{2c'(1-\Delta^t)}{k\|W_{s\star}^t\|}$. Conditioning on $F^t$, the moment generating function of the $\Delta^{t+1}$ is upper bounded by

$$E[e^{\lambda\Delta^{t+1}}|F^t] \leq \exp\{\lambda\Delta^t(1 - \frac{2c'(1-\Delta^t)}{(t+t_o)k\|W_{s\star}^t\|}) + \lambda(\eta^t)^2 B\}E\exp\{2\lambda\eta^t Z|F^t\}$$

$$= \exp\{\lambda\Delta^t(1 - \frac{\beta^t}{t+t_o}) + \lambda(\eta^t)^2 B\}E\exp\{2\lambda\eta^t Z|F^t\}$$

By Theorem 2,

$$E\{2\lambda\eta^t Z|F^t\} = 0 \text{ and } |2\lambda\eta^t Z| \leq 2\lambda\eta^t B$$

Applying Hoeffding's lemma for bounded random variable, we can bound its m.g.f. by

$$E\exp\{2\lambda\eta^t Z|F^t\} \leq \exp\{\frac{\lambda^2(\eta^t)^2 B^2}{2}\}$$

So, finally

$$E[e^{\lambda\Delta^{t+1}}|F^t] \leq \exp\{\lambda\Delta^t(1 - \frac{\beta^t}{t+t_o}) + \lambda(\eta^t)^2 B + \lambda^2\frac{(\eta^t)^2 B^2}{2}\}$$

$\square$

**Lemma 8** (Lemma from Tang & Monteleoni (2017)). *For any $\lambda > 0$,*

$$E\{e^{\lambda\Delta^t}|F^t\} \leq E\{e^{\lambda\Delta^t}|F^{t-1}\}$$

### 7.4 Technical Lemmas

**Lemma 9** (Tang & Monteleoni (2017)). *For any fixed $b \in (1, 2]$. If $C \geq \frac{b-1}{3}, \delta \leq \frac{1}{e}$, and $t \geq (\frac{3C}{b-1}\ln\frac{1}{\delta})^{\frac{2}{b-1}}$, then $t^{b-1} - 2C\ln t - C\ln\frac{1}{\delta} > 0$.*

*Proof.* Let $f(t) := t^{b-1} - 2C\ln t - C\ln\frac{1}{\delta}$. Taking derivative, we get $f'(t) = (b-1)t^{b-2} - \frac{2C}{t} \geq 0$ when $t \geq (\frac{2C}{b-1})^{\frac{1}{b-1}}$. Since $\ln\frac{1}{\delta}\frac{3C}{b-1} \geq \frac{3C}{b-1} \geq 1$, $(\ln\frac{1}{\delta}\frac{3C}{b-1})^{\frac{2}{b-1}} \geq (\frac{2C}{b-1})^{\frac{1}{b-1}}$, it suffices to show $f((\ln\frac{1}{\delta}\frac{3C}{b-1})^{\frac{2}{b-1}}) > 0$ for our statement to hold. $f((\ln\frac{1}{\delta}\frac{3C}{b-1})^{\frac{2}{b-1}}) = (\ln\frac{1}{\delta}\frac{3C}{b-1})^2 - 2C\ln\{(\ln\frac{1}{\delta}\frac{3C}{b-1})^{\frac{2}{b-1}}\} - C\ln\frac{1}{\delta} = (\ln\frac{1}{\delta})^2\frac{9C^2}{(b-1)^2} - \frac{4C}{b-1}\ln(\ln\frac{1}{\delta}\frac{3C}{b-1}) - C\ln\frac{1}{\delta} = \frac{4C}{b-1}[\frac{\frac{3}{2}C}{b-1}\ln\frac{1}{\delta} - \ln(\frac{3C}{b-1}\ln\frac{1}{\delta})] + C\ln\frac{1}{\delta}[\frac{3C}{(b-1)^2} - 1] > 0$, where the first term is greater than zero because $x - \ln(2x) > 0$ for $x > 0$, and the second term is greater than zero by our assumption on $C$. $\square$

**Lemma 10.** *Suppose our model assumptions on parameters $\epsilon, c, \alpha$ hold, and that our assumptions on the algorithmic parameter $c$ in Theorem 1 holds, then*

$$\|\epsilon\|(1 + (\alpha - \lambda)^2) \leq (1 - \frac{1}{c^2})(\frac{1-\alpha(k-1)}{k} - \alpha)$$

*Proof.*

$$(1 - \frac{1}{c^2})(\frac{1-\alpha(k-1)}{k} - \alpha)$$

$$\geq \frac{1/2 + \alpha k}{1-\alpha(k-1)}\frac{1-\alpha(k-1)}{k} - \frac{1/2\alpha + \alpha^2 k}{1-\alpha(k-1)}$$

$$= \frac{1+2\alpha k}{2k} - \frac{1/2\alpha + \alpha^2 k}{1-\alpha(k-1)}$$

$$= \frac{1}{2k} + \frac{2\alpha k(1-\alpha(k-1)) - 1/2\alpha - \alpha^2 k}{2k(1-\alpha(k-1))}$$

where

$$2\alpha k(1-\alpha(k-1)) - 1/2\alpha - \alpha^2 k$$

$$= \alpha(2k + \alpha k - 2\alpha k^2 - \frac{1}{2})$$

where the last term is greater than zero because

$$\alpha < \frac{k-1}{4k^2 - 3k + 1} < \frac{2k - 1/2}{2k^2 - k}$$

So the term

$$(1 - \frac{1}{c^2})(\frac{1 - \alpha(k-1)}{k} - \alpha) \geq \frac{1}{2k} \geq \max \|\epsilon\|(1 + (\alpha - \lambda)^2)$$

$\square$

