# OpenReview forum: "Demystifying overcomplete nonlinear auto-encoders: fast SGD convergence towards sparse representation from random initialization"
_ICLR.cc/2018/Conference — Reject_

### Official Review · AnonReviewer3 · 2017-11-09
**Limited progress, and some doubts about correctness**

**Rating:** 2
**Confidence:** 4

**Review:**

The authors study the convergence of a procedure for learning
an autoencoder with a ReLu non-linearity.  The procedure is akin
to stochastic gradient descent, with some parameters updated at
each iteration in a manner that performs optimization with respect
to the population risk.

The autoencoders that they study tie the weights of the decoder to
the weights of the encoder, which is a common practice.  There
are no bias terms in the decoder, however.  I do not see where they
motivate this restriction, and it seems to limit the usefulness of
the bias terms in the encoder.

Their analysis is with respect to a mixture model.  This is described
in the abstract as a sparse dictionary model, which it is, I guess.
They assume that the gaussians are very well separated.

The statement of Theorem says that it concerns Algorithm 1.  The
description of Algorithm 1 describes a procedure, with an
aside that describes a "version used in the analysis".

They write in the text that the rows of W^t are projected onto
a ball of radius c in each update, but this is not included
in the description of Algorithm 1.  The statement of Theorem 1
includes the condition that all rows of W^t are always equal to
c, but this may not be consistent with the updates given
in Algorithm 1.  My best guess is that they intend of
the rows of W^t to be normalized after each update (which is
different than projecting onto the ball of radius c).  This
aspect of their procedure seems restrict its applicability.

Successful initialization looks like a very strong condition to
me, something that will occur exponentially rarely, as a function
of d. (See Fact 10 of "Agnostically learning halfspaces", by Kalai, et al.)
For each row of W^*, the probability that any one row of W^o will be
close enough is exponentially small, so exponentially many rows
are needed for the probability that any row is close enough to
be, say, 1/2.  I don't see anything in the conditions of Theorem 1
that says that n is large relative to d, so it seems like its
claim includes the case where k and n are constants, like 5.
But, in this case, it seems like the claim of the probability
of successful initialization cannot be correct when d is large.

It looks like, after "successful initialization", especially
given the strong separation condition, the model as already
"got it".  In particular, the effect of the ReLUs seems to
be limited in this regime.

I have some other concerns about correctness, but I do not think
that the paper can be accepted even if they are unfounded.

The exposition is uneven.  They tell us that W^T is the transpose
of W, but do not indicate that 1_{a^t (x') > 0} is a componentwise
indicator function, and that x' 1_{a^t (x') > 0} is its
componentwise product with x' (if this is correct).

---

> ### Author Response · Authors · 2018-01-05
> **Related works added**
>
> We thank Reviewer 3 for your comments. Here are our responses to your questions/doubts.
>
> 1. Regarding your concern about bias: In our case, we especially want to include the bias term in the encoder because when used together with ReLU activation, we view it as an automatically adjusted threshold that cuts small (noisy) signals and only let pass the strongest signals: note that a neuron w_j will only be fired if w_jx+b > 0 according to ReLU activation. So negative bias -b can be viewed as controlling the level of firing threshold (which in turn controls sparsity). We have added these explanations in our updated submission.
>
> 2. Regarding your concern about the difference between the bias term used in our analysis and the one stated in Algorithm 1. First, we want to stress that the theoretical quantity we analyze can be estimated from sample (using the empirical version in Algorithm 1). In fact, the empirical version is an unbiased estimator of the theoretical quantity we analyze, just like how the stochastic gradient in SGD is an unbiased estimator of the true gradient.
> Second, the fact that our bias term can be approximated from data is already an improvement when compared to previous works (we added a Related work section in our updated version). For example, in another recent work studying ReLU activated two-layer weight-tied autoencoder (Rangamani et al, 17), the bias term is fixed to be a function of model parameters, including incoherence which is not known typically by an algorithm.
>
> 3. You guess is correct, we forgot to include the normalization step in Algorithm 1, and thanks for pointing out this bug. However, the normalization step (as discussed in the original and current version of our paper) is extremely common in deep learning. We also observed empirically that, if we do not control the norm of the weights, then when training with moderately large dictionary size, the vanilla SGD usually results in NAN weights and the training procedure becomes highly unstable. In deep learning, another common practice is to normalize the gradient (this is widely known as "gradient clipping"), which we believe have similar effect as weight normalization.
>
> In fact, we see being able to account for the normalization step in our SGD analysis as one of the strength and interesting point of our paper, because such tricks are very common in practice but lacks a theoretical justification.
>
> 4. Thanks for pointing out another bug. We did accidentally drop the dependence on dimension in our statement about success probability of random initialization. Regarding your concern about the success of initialization probability depending exponentially on d, please refer to our detailed response to Reviewer 2.

---

### Official Review · AnonReviewer2 · 2017-11-20
**Interesting work, but appears to contain a crucial bug**

**Rating:** 3
**Confidence:** 3

**Review:**

The paper considers training single-hidden-layer auto-encoders, using stochastic gradient descent, for data generated from a noisy sparse dictionary model. The main result shows that under suitable conditions, the algorithm is likely to recover the ground-truth parameters.

Although non-convex dictionary learning has been extensively studied for linear models, extending such convergence results to nonlinear models is interesting, and the result (if true) would be quite nice. Unfortunately (and unless I missed something), there appears to be a crucial bug in the argument, which requires that random initialization lead to dictionary elements sufficiently close to the ground truth. Specifically, definition 1 and lemma 1 give a bound on the success probability, which is exponentially small in the dimension d (as it should, since it essentially bounds the probability that an O(1)-norm random vector has \Omega(1) inner product with some fixed unit vector). However, the d exponent disappears when the lemma is used to prove the main theorem (bottom of pg. 10), as well as in the theorem statement, making it seem that the success probability is large. Of course, a result which holds with exponentially small probability is not very interesting. I should also say that I did not check the rest of the proof carefully.

A few relatively more minor issues:
- The paper makes the strong assumption that the data is generated from a 1-sparse dictionary model. In other words, each data point is simply a randomly-chosen dictionary element, plus zero-mean noise. With this model, dictionary learning is quite easy and could be solved directly by other methods (although I see the value of analyzing specifically the behavior of SGD on auto-encoders).
- To make things go through, the paper makes a non-trivial assumption on how the bias terms are updated (not quite according to SGD). But unless I'm missing something, a bias term isn't even needed to learn in their model, so wouldn't it be simpler and more natural to just assume that the auto-encoder doesn't have a bias term (i.e., x-> W's(Wx))?.

---

> ### Author Response · Authors · 2018-01-05
> **Exponential in dimension true, but let us explain**
>
> We thank Reviewer2 for your comment. As pointed out by Reviewer 3 as well, there was a typo in the statement of the success probability of initialization, where we dropped the exponent in d.
>
> Regarding your concern about the exponential dependence of on d, here we would like to provide three observations:
>
> 1. In dictionary learning problems, one usually preprocess the dataset in such a way that we eventually do not deal with very high dimensional data. For example, in its application to image analysis, one common way of preprocessing the dataset is to subsample random small patches from images. Each flattened patch will become training examples. And they typically have fixed dimensions (determined by the filter size) regardless of how large the original image is.
>
> 2. We are examining theoretically the performance of randomly initializing network weights by sampling the data points directly. If there is enough time, we will add the new result, as an alternative way of initialization, to this paper. Notably, the success probability do not depend exponentially in this case on the data dimension. Empirically, in fact, we also observe that initializing with random samples from the dataset works better.
>
> 3. Theoretically speaking, at least, the exponentially decaying probability due to increasing dimension can be countered by increasing the network width (also exponentially in dimension). While admittedly this is not what we observe in practice, theoretically this will work according to our current version of analysis.

---

### Official Review · AnonReviewer1 · 2017-11-28
**A potentially interesting convergence result (with errors)**

**Rating:** 2
**Confidence:** 4

**Review:**

This paper shows that an idealized version of stochastic gradient descent converges when learning autoencoders with ReLu non-linearities under strong sparsity assumptions. Convergence rates are also determined. The result is another one in the emerging line of proving convergence guarantees for non-convex optimization problems arising in machine learning, and aims to explain certain phenomena experienced in practice.

The paper is generally nicely written, providing intuitions, but there are several typos (both in the text and in the math, e.g., missing indices), which should also be corrected.

On the negative side, while the proof technique in general looks plausible, there seem to be some mistakes in the derivations, which must be corrected before the paper can be accepted. Also, the assumptions in the in the paper seem quite restrictive, and their implications are not discussed thoroughly.

The assumptions are the following:
1. The input data is coming from a mixture distribution, in the form x=w_I + eps, where {w_1,...,w_k} is a collection of unit vectors, I is uniform in {1,...,K}, eps is some noise (independent for each sample).
2. The maximum norm of the noise is O(1/k).
3. The number n of hidden neurons in the autoencoder is Omega(k) (this is not explicitly assumed but is necessary to make the probability of "incorrect" initialization small as well as the results to hold).

Under these assumptions it is shown that the weights of the autoencoder converge to the centers {w_1,...,w_k} (i.e., for any i the autoencoder has at least one weight converging to w_i). The rate of convergence depends on the coherence of the vectors w_i: the less coherent they are the faster the convergence is.

First notice that some assumptions are missing from the main statement, as the error probability delta is certainly connected to the probability of incorrect initialization: when n=1<k, the convergence result clearly cannot hold. This comes from the mistake that in Theorem 3 you state the bound for the probability P(F^\infty) instead of the conditional probability P(F^\infty|E_o) (this is present everywhere in the proof). Theorem 3 should also depend on delta_o, which is used in the definition of F^\infty.

Theorem 2 also seems incorrect. Intuitively, the question is why it cannot happen that two neurons contribute to reproducing a given w_i, and so neither of their weights converge to w_i: E.g., assuming that {w_1,...,w_k,w_1',...,w_k'} form an orthogonal system and the noise is 0, the weight matrix of size n=2k defined as W_{2i-1,*}^T = 1/sqrt{2}(w_i + w'_i) and W_{2i,*}^T=1/sqrt{2}(w_i - w'_i), i \in [k], with 0 bias can exactly recover any x=w_i (indeed, W_{2j-1,*} x= W_{2j,*} x = 1/sqrt{2}, while the other products are 0, and so W^T W x = W^T W w_j = 1/sqrt{2}(W_{2j-1,*}+W_{2j,*})^T = w_j). Then SGD does not change the weights and hence cannot recover the original weights {w_i }, in particular, it cannot increase the coherence in any step, contradicting Theorem 2. This counterexample can be extended even to the situation when k>d, as--in fact--we only need that the existence of a single j such that w_j and w'_j are orthogonal and also orthogonal to the other basis vectors.

The assumptions are also very strange in the sense that the norm of the noise is bounded by O(1/k), thus the more modes the input distribution has the more separable they become. What motivates this scaling? Furthermore, the parameters of the algorithm for which the convergence is claimed heavily depend on the problem parameters, which are not known. How can you instantiate the algorithm then (accepting the ideal definition of b)? What are the consequences?

Given the above, at this point I cannot recommend the paper for acceptance. However, if the above problems are resolved, I would be very happy to see the paper at the conference.


Other comments
-----------------------
- Add a short derivation why the weights of the autoencoder should converge to the w_i.
- Definition 3: C_j is not defined in the main text.
- While it is mentioned multiple times that the interesting regime is d<n, this is actually never used, nor needed (personally, I have never seen such an autoencoder--please give some references). What is really needed is n>k, which is natural if one wants to preserve the information, and also k>d for a rich family of distributions.
- The area of the spherical cap is well understood (up to multiplicative constants), and better bounds than yours are readily available: with a cap of height 1-t, for sqrt{2/d}<t<1, the relative surface of the cap is between P/6 and P/2 where
P=1/(t \sqrt{d}) (1-t^2)^{(d-1)/2}; see, e.g., A. Brieden, P. Gritzmann, R. Kannan, V. Klee, L. Lovasz, and M. Simonovits. Deterministic and randomized polynomial-time approximation of radii. Mathematika. A Journal of Pure and Applied Mathematics, 48(1-2):63–105, 2001.
- The notation section should be brought forward (or referred the fist time the notation is actually used).
- Instead of unit spherical Gaussian you could simply say uniform distribution on the unit sphere
- While Algorithm 1 is called "norm-controlled SGD training," it does not control the norm at all.

---

> ### Author Response · Authors · 2018-01-05
> **Corrected typos, and let us clarify some misunderstanding you had about our analysis**
>
> We are grateful for your careful examination of our paper. We have already corrected several typos in our statements (we haven't found mistakes in our proofs), simplified our analysis and parameters. Below are some of our clarification which we hope can help clear some doubts you had about our analysis.
>
> 1. delta is included in the statement of Theorem 1 (our main theorem); it is in fact not related to the probability of successful initialization, but a handy parameter to help us control the probability of martingale convergence in the later stage of the algorithm (addressed in Theorem 3)
>
> 2. Regarding your question about Theorem 3, the event F^{\infty} in fact implies E^o, so with or without conditioning on E^o, the probability is the same.
>
> 3. For Theorem 2, it can happen that "two neurons contribute to reproducing a given w_i" (in fact, this is exactly what we want by adding more neurons and is beneficial as revealed by our analysis). But in your example, it will happen that one neuron will contribute equally to reproducing two different ground-truth dictionary items; this will not happen under our definition of "active" neurons by the "unique firing condition", which a prerequisite to proving Theorem 2.
>
> Essentially, unique firing condition is guarantee by our definition of successful initialization (the inner product between any dictionary item and at least one of the (normalized) neuron is required to be strictly larger than 1/sqrt(2).
> So neurons taking the specific values given in your example are considered effectively "dead" in our analysis (g(s)=0).
> Your example perhaps also illustrates why the bias term is beneficial to keep; the bias will serve as a threshold to filter out neurons that are close to the "decision boundary" and not specializing to learning a single dictionary item.
>
> 4. We can provide some rough intuition as for why the norm of noise depends inversely on k, i.e., the true number of dictionary items; the coherence-to-noise ratio can be viewed as the "signal-to-noise" ratio of our model.  Intuitively, noise cannot scale larger than signal (coherence). In sparse dictionary learning models, coherence usually scales inversely with the number of dictionary items. The typical scale is coherence=1/sqrt(k), while in our case it is 1/k, which is admittedly worse. However, previous theoretical guarantee usually needs to know the exact value of coherence (e.g., see Rangamani et al cited in the updated version of our paper) and thus sets the threholding parameter using this knowledge, while in our case we automatically adjust the bias term using data (which is more practical than the theoretical thresholding method).
>
> Also, the norm of the noise does not actually have to be bounded in a deterministic sense. Thanks for pointing this out. We added a footnote to explain this in our paper: we can relax and assume that the noise has, e.g., sub-Gaussian tails.
>
> 5. While proving the success of our SGD variant depend on knowing either upper or lower bounds on certain model parameters, we do not "heavily" depending on them. This is especially true in the updated version of our paper, where there are only two model parameters k and \lambda left, representing the number of true dictionary items and the incoherence.
> For k, we only need a loose lower bound (as part of setting our norm parameter). In contrast, for almost all clustering problems, for example, the number of clusters which corresponds to the number of dictionary items in this case, needs to be known exactly.
> For \lambda, we only need an upper bound to set our learning rate parameter.
> The bias update we use can also be well approximated by sampling the data. In contrast, e.g., the recent related work of Rangamani et al, who also studies ReLu activated overcomplete autoencoders, sets their bias term using the exact knowledge of incoherence.
>
> In our updated paper, we added a Related work section, and added more discussion motivating why the case n>d is relevant (this is in fact known as "overcomplete" dictionary learning in the literature). We hope our added explanations and discussions can help you better appreciate our paper.

---

### Author Response · Authors · 2018-01-06
**Overview of our responses and updated submission**

We thank all reviewers for spending your time in reviewing our work and providing insightful feedbacks.

Admittedly, our first submission was under a very limited time budget and hence bug-prone. We have corrected several bugs/typos mentioned by reviewers and beyond.

However, after repeatedly going through our proofs, we have not found any fatal mistake in the proof. Furthermore, we have devoted a lot of time in greatly simplifying our analysis. As a result,

1. We were able to eliminate some model parameters and hard-to-parse interdependence between parameters. Now, our results are stated in a much more crisp way.

2. Regarding the motivation of studying over-complete dictionary learning,  we also added reference and discussion in the paper. Moreover, we added a Related work section to discuss recent advances in studying two-layer auto-encoders, and compare our results with existing ones.

3. Regarding your concern about the success probability being exponentially dependent on dimension, we have provided three explanations under our response to Reviewer 2's comments. Since our submission, we have also explored and added performance guarantee of another form of random initialization, initializing by randomly sampling data points. With data initialization, we are able to provide a much stronger and realistic guarantee on the success probability: if the network width increases of order at least \Omega(k^3), then with high probability, successful initialization can be guaranteed. In contrast, with Gaussian initialization, we need network width to grow as \Omega(k^d).

Thank you

---

### Decision · Program_Chairs · 2018-01-29
**ICLR 2018 Conference Acceptance Decision**

**Decision:**

Reject

**Comment:**

The reviewers have unanimously expressed strong concerns about the technical correctness of the theoretical results in the paper. The paper should be carefully revised and checked for technical errors. In its current form, the paper is not suitable for acceptance at ICLR 2018.